# Immuno-protective vesicle-crosslinked hydrogel for allogenic transplantation

Yuqian Wang[1], Renqi Huang[1], Yougong Lu[1], Mingqi Liu[1] & Ran Mo [1] ✉

The longevity of grafts remains a major challenge in allogeneic transplantation due to immune rejection. Systemic immunosuppression can impair graft function and can also cause severe adverse effects. Here, we report a local immuno-protective strategy to enhance post-transplant persistence of allografts using a mesenchymal stem cell membrane-derived vesicle (MMV)-crosslinked hydrogel (MMV-Gel). MMVs are engineered to upregulate expression of Fas ligand (FasL) and programmed death ligand 1 (PD-L1). The MMVs are retained within the hydrogel by crosslinking. The immuno-protective microenvironment of the hydrogel protects allografts by presenting FasL and PD-L1. The binding of these ligands to T effector cells, the dominant contributors to graft destruction and rejection, results in apoptosis of T effector cells and generation of regulatory T cells. We demonstrate that implantation with MMV-Gel prolongs the survival and function of grafts in mouse models of allogeneic pancreatic islet cells and skin transplantation.

Transplantation is a lifesaving therapy for end-stage failure or permanent loss of organs by engraftment of cells, tissues or organs with the aim of repairing and restoring essential functions[1]. For example, cell transplantation using hematopoietic stem cells is the only curative therapeutic modality for fatal hematologic malignancies that are highly resistant to chemotherapy at present[2,3]. Pancreatic islet transplant is an effective clinical treatment of β-cell replacement to attain glycaemic control for the patients with type 1 diabetes mellitus[4,5]. Skin grafting as a form of tissue transplantation is the operative gold standard for treatment of burn injuries[6] as well as the most common choice for treatment of chronic non-healing wounds[7]. Solid organ transplantation still remains integral to treat irreversible diseases of liver, kidney, lung and heart that no longer function normally[8–11].

Transplantation is categorized into autologous, syngeneic, allogeneic and xenogeneic transplantation depending on the genetic disparity between donor and recipient[12]. Allogeneic transplantation that refers to grafting to a recipient from a non-genetically-identical donor of the same species is the most widely used in clinic due to reduced time to treatment, ability to standardize product, low production cost, and no additional invasive procedure required for recipient. However, the immune system represents a significant obstacle to allogeneic transplantation[13], which identifies allografts as foreign and elicits immune rejection that would destroy the transplanted grafts ultimately. In patients with type 1 diabetes, transplanted islets are also targeted by the recurrence of autoimmunity which further limits long-term graft survival. Immunosuppressive drugs such as rapamycin (RAPA) and tacrolimus are commonly applied for prevention and treatment of the allograft rejection by inhibiting the systemic immune response[14,15]. Nevertheless, repeated administration of nonspecific immunosuppressants disrupts immune homeostasis, leading to increased risks of organ toxicity, myelosuppression, infections and malignancy[16,17]. Cellular therapy is recognized as a promising alternative to classical immunosuppressants for achieving allograft acceptance[18,19]. Mesenchymal stem cells (MSCs) possessing immunomodulatory properties have been approved for treatment of graft-versus-host disease and increasingly explored for treatment of various autoimmune diseases[20,21]. Costimulatory molecules like FasL, PD-L1 and ICAM1 on the cell surface of MSCs can induce cell death and cell cycle arrest of T effector (Teff) cells that are the ringleader for allograft destruction and rejection, and increase regulatory T (Treg) cells by interacting with the corresponding receptors on the Teff cells[22,23]. Such MSC-to-T cell contact effects attenuate immune responses and

[1]State Key Laboratory of Natural Medicines, Jiangsu Key Laboratory of Drug Discovery for Metabolic Diseases and Jiangsu Key Laboratory of Drug Design and Optimization, Center of Advanced Pharmaceuticals and Biomaterials, School of Life Science and Technology, China Pharmaceutical University, Nanjing 211198, China. ✉e-mail: rmo@cpu.edu.cn

promote allograft acceptability. Local co-transplantation with MSCs can delay the immune rejection and improve the alloislet survival[24,25]. However, MSCs have the risk of tumorigenicity due to uncontrollable self-renewal and multidirectional differentiation, and the lack of persistent immunomodulatory activity resulting from limited local retention, aging and eventual death after transplant[26,27].

In this work, we report a local immuno-protective strategy using an MSC-derived vesicles-crosslinked hydrogel that serves as an immunosuppressive "niche" for acceptance of allogeneic transplantation (Fig. 1). MSC membrane vesicles (denoted MMVs) are obtained through membrane extraction and extrusion, and naturally inherit the surface proteins expressed on MSCs including FasL. Interferon (IFN)-γ is employed to stimulate upregulation of the PD-L1 expression on MSCs[28]. The resulting MMVs are anchored with the thiol-terminated PEGylated phosphatidylethanolamine (DSPE-PEG-SH) to be a nanovesicular crosslinker. Hyaluronic acid (HA), a biocompatible natural polysaccharide is used for hydrogel matrix[29] and modified with the thiol-reactable acrylated pendants[30]. The MMV-crosslinked hydrogel (designated MMV-Gel) is fabricated by simple mixing of the thiol-decorated MMVs and the acrylated HA (designated a-HA) via a Michael addition reaction[31] (Fig. 1a). We hypothesize that the obtained MMV-Gel provides a local immuno-protective microenvironment to achieve sustained survival and function of the transplanted allografts without chronic systemic immunosuppression. To validate our assumption, MMV-Gel is employed on the transplantation models of allogeneic islet and skin. After local implantation, MMV-Gel prolongs the retention of MMVs at the transplanted site. MMVs are homogeneously and firmly tethered in the hydrogel network, minimising the risk of systemic leakage and achieving persistent immunomodulatory effect. MMV-Gel also improves the long-term preservation of the graft to maintain its function (Fig. 1b). When the allograft is transplanted, the host immune system elicits immunologic response by recognizing alloantigens. Upon activation, Teff cells, the critical immunologic executer infiltrate into the transplanted site with a message to attack allograft, which upregulate the expression of Fas and programmed cell death protein 1 (PD1)[32,33]. FasL presenting on MMVs binds to Fas on the Teff cells, triggering the apoptosis of Teff cells[34]. Meanwhile, PD-L1 upregulatedly expressed on MMVs is coupled with PD1 on the Teff cells, promoting generation of Treg cells for suppressing the activity of Teff cells[25]. In addition to presentation of FasL and PD-L1, MMVs also harbour a collection of other immunomodulatory surface biomolecules[22], such as VCAM1, VTCN1 and CXCL10, which enable MMV-Gel to be an immunosuppressive protein compounding depot to reconstruct a localized immuno-protective microenvironment tilted towards the Treg cells for acceptance of allogeneic transplantation. We show that the MMV-Gel-based immuno-protection enhances the survival and function of grafts in the mouse models of allogeneic islet and skin transplantation.

## Results

### MMV-Gel induces apoptosis and inhibits proliferation of the activated T cells

MSCs were isolated from the bone marrow of the BALB/c mice[35]. The purity of MSCs was characterized by positive CD29 and CD90 expression with negative CD34 and CD45 expression (Supplementary Fig. 1). We analysed the surface expression of FasL on MSCs, red blood cells (RBCs), platelets (PLTs) and activated platelets (aPLTs) (Fig. 2a). FasL was rarely or lowly expressed on RBCs and PLTs. Thrombin-induced activation brought about elevated expression of FasL on PLTs[36]. In stark contrast, MSCs showed significantly higher surface expression of FasL. The FasL expression of MSCs was about 366-fold relative to that of aPLTs. Furthermore, we investigated the effect of

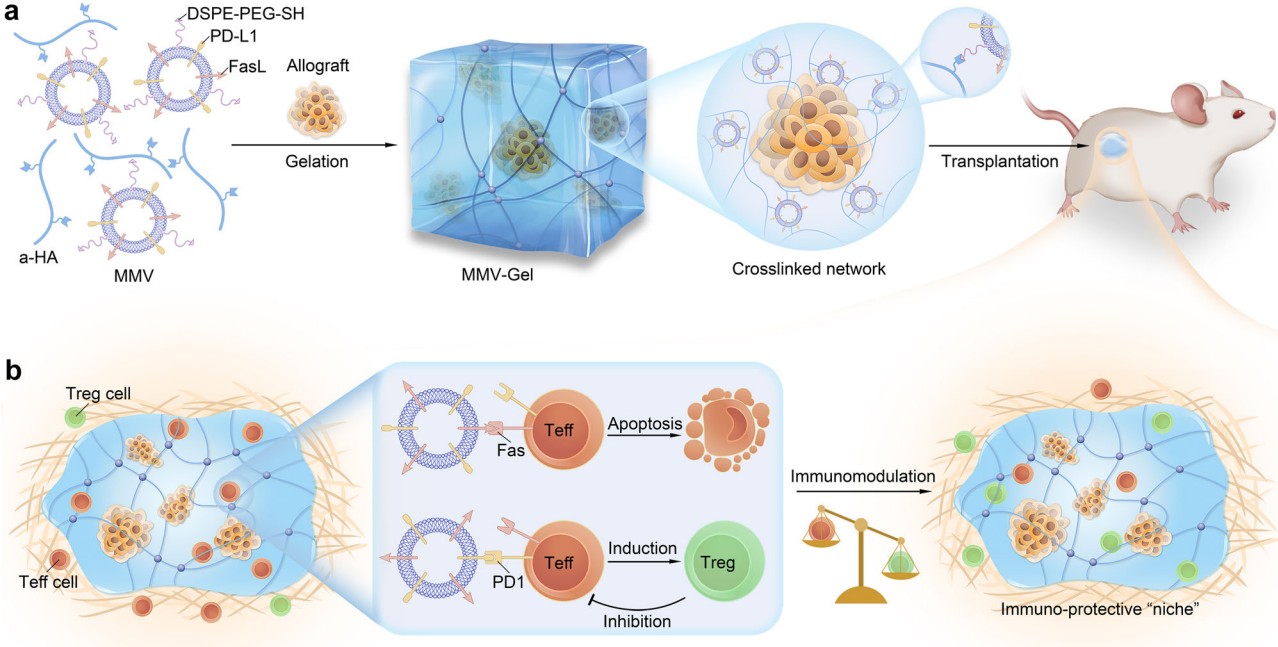

**Fig. 1 | Schematic of local immuno-protective niche implemented by MMV-Gel for prolonged survival and function of allogeneic transplants. a** Schematic of preparation of MMV-Gel for local delivery of allografts. MMV-Gel is formed by mixing thiol-modified FasL/PD-L1-expressing MMVs and a-HA via specific thiol-acrylate Michael addition reaction. As an example, the transplanted grafts can be embedded in the crosslinked network of MMV-Gel. **b** Schematic of mechanism of MMV-Gel serving as an immuno-protective niche to achieve allograft acceptance. MMV-Gel holds the tethered MMVs to enhance their retention at the implanted site. The local persistence of MMV-Gel improves long-term preservation of the graft. Upon acute immune response to allotransplantation, the Teff cells increasingly infiltrate into the transplanted region with the mission to attack and destruct grafts, ultimately leading to graft rejection, which are characterized to highly express Fas and PD1 following activation. MMV-Gel provides an immuno-protective microenvironment to resist the Teff cell-mediated immune response by the MMV-presenting FasL and PD-L1 binding to their specific receptors, Fas and PD1 on the Teff cells to induce apoptosis of Teff cells and elevation of Treg cells.

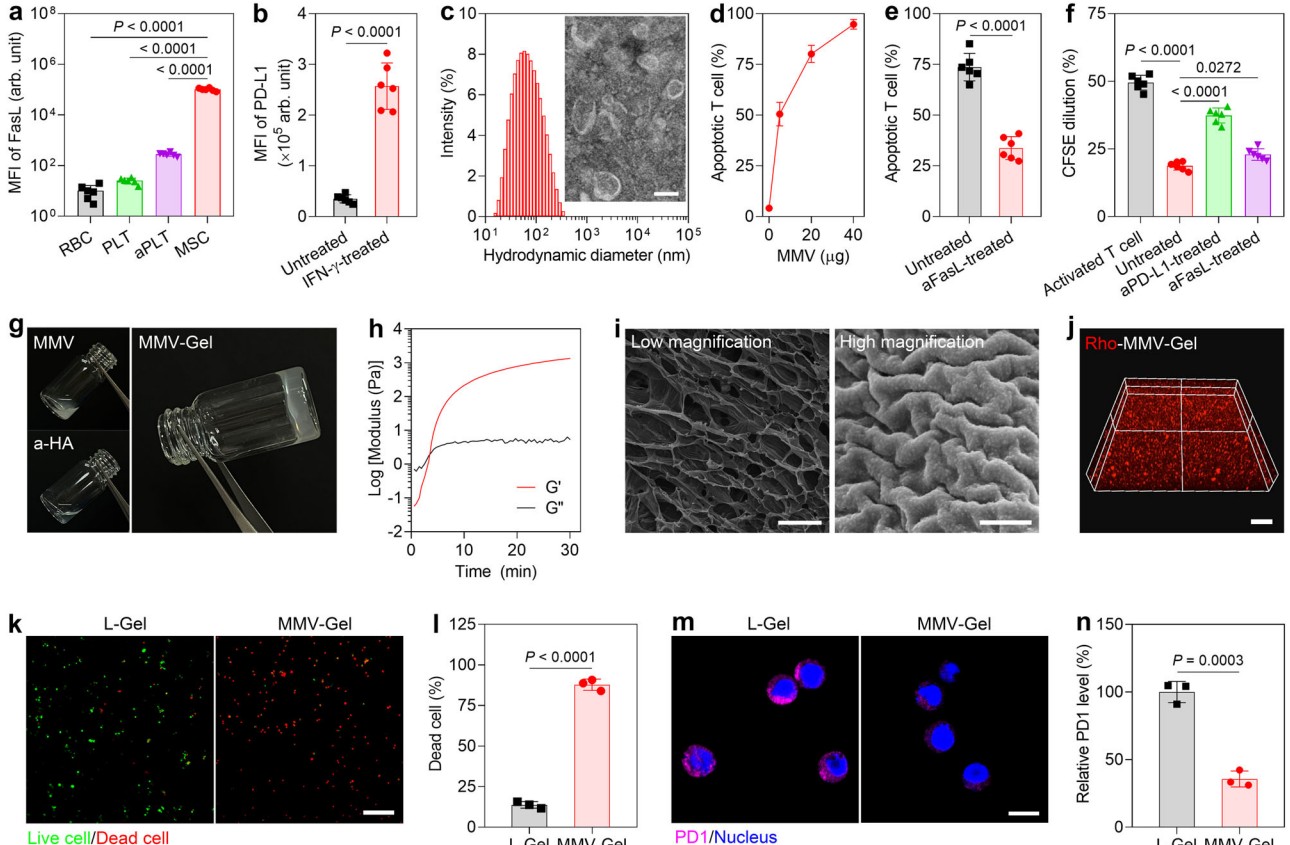

**Fig. 2 | MMV-Gel triggers apoptosis and suppresses proliferation on the activated T cells. a** Expression of FasL on RBCs, PLTs, aPLTs and MSCs determined by flow cytometry. MFI, mean fluorescent intensity. arb. unit, arbitrary unit. **b** Expression of PD-L1 on MSCs after treatment with IFN-γ for 12 h determined by flow cytometry. **c** Particle size and TEM image of MMVs. Scale bar, 50 nm. **d** Total (early plus late) apoptotic percentages of T cells within the activated splenocytes after treatment with varying amounts of MMVs determined by Annexin V-FITC/PI double-staining assay. **e** Total apoptotic percentages of T cells within the activated splenocytes after treatment with MMVs (20 μg) in the absence and presence of aFasL. **f** Proliferation of T cells within the activated splenocytes after treatment with MMVs (10 μg) in the absence and presence of aPD-L1 or aFasL determined by CFSE dilution assay. **g** Gelation of MMV-Gel by mixing MMVs and a-HA examined by tube inversion assay. **h** Rheology measurement of MMV-Gel. **i** SEM images of MMV-

Gel. Scale bars, 100 μm (low magnification) and 500 nm (high magnification). **j** Three-dimensional reconstructed confocal microscopic image of Rho-MMV-Gel. Scale bar, 50 μm. Fluorescent imaging (**k**) and quantification (**l**) of viability of the activated T cells infiltrating in L-Gel and MMV-Gel for 48 h examined by calcein AM/PI double-staining assay. Scale bar, 100 μm. Fluorescent imaging (**m**) and quantification (**n**) of expression of PD1 on the activated T infiltrating in L-Gel and MMV-Gel for 24 h. Scale bar, 5 μm. Representative is displayed from 3 independent experiments (**c, g–k, m**). Data are shown as mean ± standard deviation (s.d.) (*n* = 6 independent samples in **a, b, d–f**; *n* = 3 independent samples in **l, n**). One-way ANOVA with Tukey post-hoc test was used for statistical analysis of **a, f**. Two-tailed unpaired *t*-test was used for statistical analysis of **b, e, l, n**). Source data are provided as a source data file.

IFN-γ on the PD-L1 expression on MSCs. The IFN-γ-treated MSCs exhibited a concentration-dependent increase of PD-L1 expression (Supplementary Fig. 2a). The PD-L1 expression of MSCs treated with IFN-γ at an optimized concentration of 1 ng/mL for 12 h was 7.4-fold that of the untreated MSCs (Fig. 2b). The IFN-γ treatment did not cause any noticeable influence on the expression of FasL at the studied concentration (Supplementary Fig. 2b).

Cell membranes were extracted from the IFN-γ-treated MSCs using the differential centrifugation method, and the corresponding MMVs were prepared by gentle sonication and repeated extrusion[37]. The obtained MMVs had a hydrodynamic diameter of 60 nm (Fig. 2c) and a zeta potential of −40 mV (Supplementary Fig. 3). As observed by transmission electron microscopy (TEM), MMVs had an apparently ring-like vesicular morphology (Fig. 2c). The SDS-PAGE analysis elucidated that MMVs preserved cell membrane proteins of MSCs (Supplementary Fig. 4a). Meanwhile, MMVs harbour high levels of both FasL and PD-L1 that were comparable to MSCs, as substantiated by western blotting analysis (Supplementary Fig. 4b) and ELISA assay (Supplementary Fig. 4c, d). Of note, MMVs showed marked prolonged persistence of PD-L1 expression on the surface compared with MSCs (Supplementary Fig. 5).

We monitored the changes in the expression of Fas and PD1 on the T cells following activation. The splenocytes were activated by concanavalin A (Con A), and the expression of Fas and PD1 on the T cells within the activated splenocytes were analysed. The activated T cells showed elevated surface expression of both Fas and PD1 (Supplementary Fig. 6). We evaluated the effect of MMVs on inducing apoptosis of the activated T cells. The T cells within the activated splenocytes exhibited significantly increased apoptotic percentage after treatment with MMVs in a dose-dependent manner (Fig. 2d and Supplementary Fig. 7). A significant reduction of apoptotic percentage of T cells was determined after treatment with MMVs in the presence of anti-FasL antibody (aFasL) (Fig. 2e and Supplementary Fig. 8). We further assessed the capacity of MMVs to suppress proliferation of the activated T cells using carboxyfluorescein succinimidyl ester (CFSE) dilution assay. MMVs displayed a strong proliferative inhibition on the T cells within the activated splenocytes, and higher feeding amount of MMVs produced stronger anti-proliferative effect (Supplementary Fig. 9). Pre-treatment with anti-PD-L1 antibody (aPD-L1) significantly restored the proliferation of the activated T cells (Fig. 2f and Supplementary Fig. 10). The FasL blockade caused partial recovery of the activated T cell proliferation. Next, we characterized the T phenotype

of the Con A-treated splenocytes after incubation with MMVs. The proportions of both CD8[+] Teff (CD8[+]CD44[+]CD62L[−]) and CD4[+] Teff (CD4[+]CD44[+]CD62L[−]) cells in the activated splenocytes significantly reduced (Supplementary Fig. 11a, b), while the percentage of Treg (CD4[+]FoxP3[+]) cells markedly increased (Supplementary Fig. 11c). These results suggest that the obtained MMVs with high surface expressions of biologically active FasL and PD-L1 can effectively trigger apoptosis and suppress proliferation of the activated T cells by the Fas/FasL and PD1/PD-L1 signalling pathways.

MMV-Gel was constructed by simply mixing the separate solution containing MMVs and a-HA. MMVs were modified with the reactive thiol groups by DSPE-PEG-SH via its hydrophobic alkyl chain embedded into the lipid bilayered membrane. The optimized ratio of MMVs to DSPE-PEG-SH was about 1:3 (mg protein: mg) at the a-HA concentration of 50 mg/mL (Supplementary Fig. 12). The thiol group on MMVs could selectively react with the acryl groups on a-HA (Supplementary Fig. 13) at pH 7.4 through a Michael addition reaction to trigger gelation for formation of MMV-Gel, a mild reaction without additional chemical crosslinkers (Fig. 2g). After treatment with a high concentration of Triton X-100 that can disrupt the lipid membrane, MMV-Gel underwent a phase transition from a gel-like (semisolid) state to a liquid-like (fluid) state (Supplementary Fig. 14), indicating that the MMV-mediated crosslinking drives the gelation. The changes of characteristic proton peaks before and after hydrogel formation were detected by [1]H NMR spectrum. The absence of proton peaks at $\delta$ 6.00–6.44 ppm (acrylic acid protons)[38] in MMV-Gel validated that all the acryl groups of a-HA had reacted with the thiol groups on MMVs (Supplementary Fig. 15). The rheological evaluation was performed on the gelation kinetics of MMV-Gel. Elastic modulus (G′) rapidly ascended over time after addition of MMVs, and eventually reached a plateau far above viscous modulus (G″) within 30 min (Fig. 2h), which was solid evidence of hydrogel formation[39]. Scanning electron microscope (SEM) images exhibited that crosslinked network supports the microscale structure of hydrogel in which a number of MMVs were visible (Fig. 2i). Three-dimensional scanning confocal microscopic image also showed that the rhodamine-labelled MMVs (Rho-MMVs) were uniformly distributed in the hydrogel (Fig. 2j).

We further evaluated the capability of MMV-Gel as an immuno-suppressive depot to induce the activated T cell death in vitro. After sorted from splenocytes, the T cells were stimulated by Con A. The activated T cells were seeded on the hydrogel. After 48 h of incubation, the viabilities of T cells infiltrating within the hydrogel were examined by calcein AM/PI double staining. Confocal microscopic imaging showed that a large quantity of cells was alive within the control hydrogel formed by the thiol-modified liposome and a-HA (designated as L-Gel), while massive dead cells were observed in MMV-Gel (Fig. 2k). The dead cell ratio of the activated T cells within MMV-Gel was significantly higher than that in L-Gel (Fig. 2l). These data suggest that FasL anchored on MMV binding with Fas overexpressed on the activated T cells results in efficient induction of the infiltrating T cell death within MMV-Gel. In addition, we also evaluated the efficiency of MMV-mediated PD-L1 binding to PD1 of the infiltrating T cells within MMV-Gel. After seeded on the hydrogel surface for 24 h, the activated T cells infiltrating in the hydrogel were stained by fluorescence-labelled anti-PD1 antibody, followed by confocal microscopic observation. The fluorescent signal on the surface of the cells in MMV-Gel was detected to be significantly lower than that within L-Gel (Fig. 2m), which is due to the competitive binding of the MMV-expressing PD-L1 and the used anti-PD1 antibody to PD1 on the activated T cells. MMV-Gel presented the PD-L1/PD1 binding efficiency of about 64% relative to L-Gel (Fig. 2n). The results indicate that MMV-Gel supports efficient binding of the anchored PD-L1 on MMVs to PD1 expressed on the activated T cells, which lays a substantial foundation for induction of Treg cell formation.

## MMV-Gel offers an immunosuppressive microenvironment to protect islet allograft

Murine pancreatic islets were isolated from the donor C57BL/6 mouse. The islet-embedded MMV-Gel (designated as islet/MMV-Gel) was obtained by incubating the a-HA solution containing islets with MMVs for 1 h. Cryo-SEM and three-dimensional scanning confocal imaging confirmed that the islets were successfully encapsulated within the interwoven network of hydrogel (Fig. 3a) and MMVs were distributed around the islets (Supplementary Fig. 16). The immunofluorescent staining showed positive expression of both insulin and glucagon of the islet in MMV-Gel (Fig. 3b). The static glucose-stimulated insulin secretion (GSIS) assay[40] illustrated that the islets cultured in MMV-Gel maintained insulin secretion by responding to variation of environmental glucose levels (Fig. 3c). The corresponding calculated insulin stimulation index showed no statistical difference between islet/MMV-Gel and free islet (Fig. 3d). The ex vivo insulin secretion capacity of islet/MMV-Gel was further estimated within one week. No significant change of insulin secretion from islet/MMV-Gel within 3 d, while the decline in the insulin secretion after 7 d was ascribed to the natural attenuation of the islet function under the condition of ex vivo culture (Fig. 3e, f and Supplementary Fig. 17). In addition, the islets in MMV-Gel exhibited a high viability (Supplementary Fig. 18). Meanwhile, the secretion of pro-inflammatory cytokines[41], such as MIP-1α, MCP-1 and IL-6, was not statistically different for the islet/MMV-Gel and free islet groups within one week of culture (Supplementary Fig. 19). These results suggest that MMV-Gel presenting FasL and PD-L1 acts as a local depot for islet transplantation without affecting viability and function of the islet payloads.

Next, we evaluated the immuno-protective function of MMV-Gel on the islet allograft. The hydrogel embedding the islets (C57BL/6) was incubated with the activated T cells (BALB/c) for 48 h. The islet viability was first examined by acridine orange/PI double staining. The islets in MMV-Gel showed noticeably higher viability with significantly lower dead cell percentage than that in L-Gel (Fig. 3g, h), evidencing that MMV-Gel defends the encapsulated islets against the immune attack and apoptosis induction by the activated T cells. The data acquired from the static GSIS assay further showed that the insulin secretion level of islet/MMV-Gel was significantly higher than that of islet/L-Gel exposed to either low or high concentration of glucose (Fig. 3i). islet/MMV-Gel presented substantially higher stimulation index than islet/L-Gel (Fig. 3j), evidencing that the islets preserve the glucose-responsive insulin-secretory function against the T cell-mediated immune attack under protection of MMV-Gel. Taken together, MMV-Gel provides potent immuno-protection for the allogeneic islets by preventing viability decay and function impairment from the activated T cells.

To validate prolonged retention of MMVs at the transplantation site with the assistance of the hydrogel, DiD-labelled MMV-Gel (DiD-MMV-Gel) was prepared and implanted under the left kidney capsule of mice, and the variation of the DiD fluorescent signal was monitored (Supplementary Fig. 20a). The DiD signal in the mice receiving either DiD-MMV or DiD-MMV/HA (the mixture of DiD-MMV and HA) as control groups rapidly reduced within 7 d and nearly totally disappeared after 14 d. In sharp contrast, the DiD fluorescent signal of the mice with local implantation of DiD-MMV-Gel persisted within the transplanted kidney up to 28 d. At 28 d post-implantation, the kidney and other normal major tissues were harvested from the mice for ex vivo imaging. Strong DiD signal was observed in the implanted kidney, while no signal was detectable in other tissues in the DiD-MMV-Gel group (Supplementary Fig. 20b). Accordingly, the crosslinked hydrogel prolongs the retention of MMVs with the immunoregulatory proteins, FasL/PD-L1 in the transplanted region while minimizing the risk of their systemic effects.

The in vivo immuno-protective potential of MMV-Gel for allogeneic islet transplantation was assessed on the streptozotocin-induced diabetic mouse model[42]. MMV-Gel encapsulating the islets

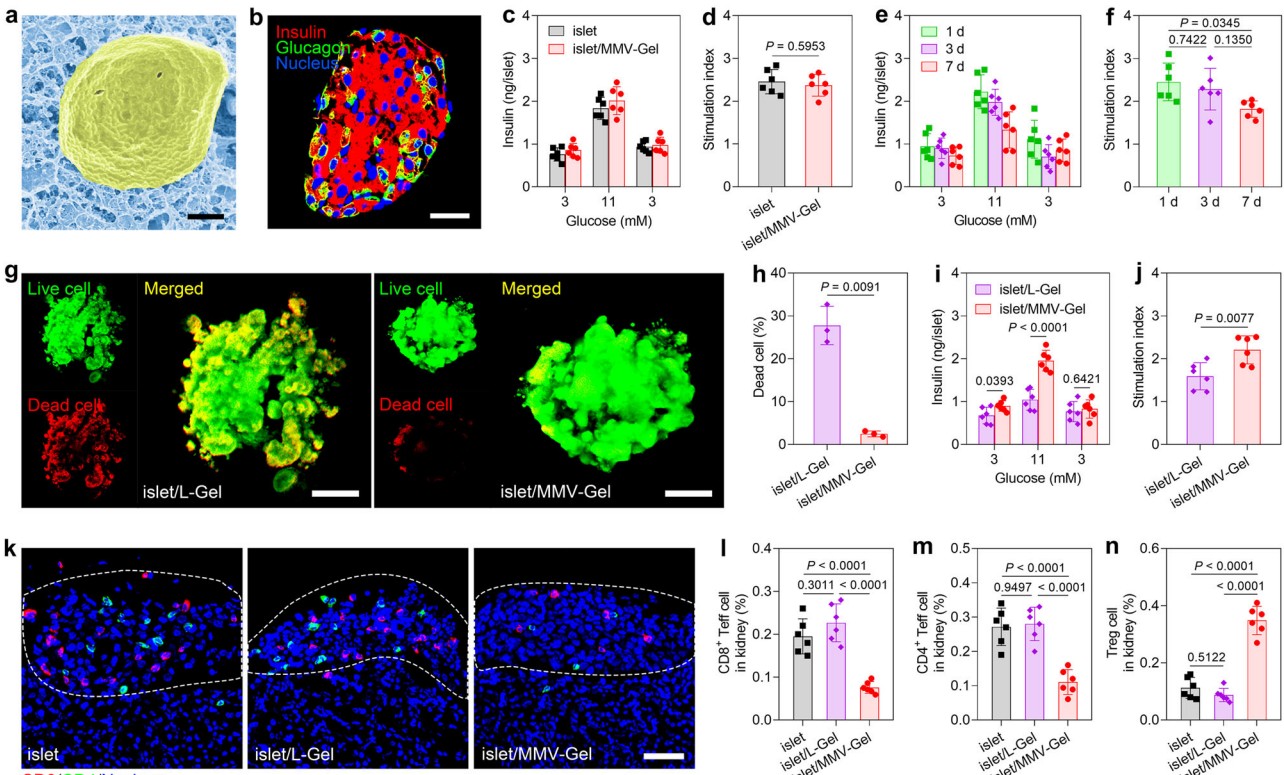

**Fig. 3 | MMV-Gel protects the encapsulated islet allograft against the activated T cell. a** Cryo-SEM image of islet/MMV-Gel. Pseudocolour processing was used for image display of islet in yellow and hydrogel matrix in blue. Scale bar, 10 μm. **b** Expression of insulin and glucagon in islet/MMV-Gel examined by immunofluorescent staining. Scale bar, 25 μm. GSIS (**c**) and stimulation index (**d**) of free islet and islet/MMV-Gel. GSIS (**e**) and stimulation index (**f**) of islet/MMV-Gel within 7 d. Fluorescent imaging (**g**) and quantification (**h**) of viability of islet/L-Gel and islet/MMV-Gel after incubation with the activated T cells for 48 h examined by acridine orange/PI double-staining assay. Scale bar, 25 μm. GSIS (**i**) and stimulation index (**j**) of islet/L-Gel and islet/MMV-Gel after incubation with the activated T cells for 48 h. **k** Infiltration of CD8+ and CD4+ T cells in the transplanted site of kidney at 7 d post-transplantation of islet, islet/L-Gel and islet/MMV-Gel examined by immuno-fluorescent staining. White dotted lines indicate the graft. Scale bar, 50 μm. Proportions of CD8+ Teff (CD8+CD44+CD62L−) (**l**), CD4+ Teff (CD4+CD44+CD62L−) (**m**) and Treg (CD4+FoxP3+) (**n**) cells in kidney at 7 d post-transplantation of islet, islet/L-Gel and islet/MMV-Gel determined by flow cytometry. Representative is displayed from 3 independent experiments (**a**, **b**, **g**, **k**). Data are shown as mean ± s.d. (*n* = 6 independent samples in **c**–**f**, **i**, **j**; *n* = 3 independent samples in **h**; *n* = 6 mice in **l**–**n**). Two-tailed unpaired *t*-test was used for statistical analysis of **d**, **h**–**j**. One-way ANOVA with Tukey post-hoc test was used for statistical analysis of **f**, **l**–**n**. Source data are provided as a source data file.

from the donor C57BL/6 mouse was implanted under the left kidney capsule of the recipient BALB/c mouse. After 7 d, the infiltrating T cells within the transplant region of the kidney were examined by immunofluorescent staining and analysed by flow cytometry. Grafting of allogeneic islets evoked immune rejection response as presented by massive CD8+ and CD4+ Teff cells infiltrating into the transplantation area (Fig. 3k). The application of L-Gel did not alleviate local immune response. No significant difference in the intrarenal proportion of Teff cells (CD8+CD44+CD62L− and CD4+CD44+CD62L−) for the free islet and islet/L-Gel groups (Fig. 3l, m and Supplementary Fig. 21). Encouragingly, the frequency of Teff cell infiltration was significantly lower in the kidney transplanted with islet/MMV-Gel. Of note, transplantation of islet/MMV-Gel also showed significantly higher percentage of Treg cells (CD4+FoxP3+) within the kidney compared with implantation of either free islet or islet/L-Gel (Fig. 3n and Supplementary Fig. 21). Moreover, the proportion of apoptotic Teff (CD8+CD44+CD62L−TUNEL+ and CD4+CD44+CD62L−TUNEL+) and induced Treg (iTreg) cells (CD4+FoxP3+Nrp-1−) was significantly higher in the kidney that received the islet/MMV-Gel treatment (Supplementary Figs 22, 23). The data manifest that MMV-Gel protects the islet allograft against immune rejection by reducing the accumulation of Teff cells by triggering their apoptosis and increasing the distribution of Treg cells by promoting their generation at the transplantation site. Furthermore, the Teff and Treg cells in the kidney-draining lymph node (KDLN) and spleen were analysed at 7 d post-transplantation (Supplementary Figs 24, 25).

Percentages of Teff cells were significantly reduced in both KDLN and spleen of the islet/MMV-Gel-transplanted mice compared with that of either free islet- or islet/L-Gel-transplanted mice. The spleen revealed no statistical differences in the Treg cell proportions among different treatments, while the proportion of Treg cells was notably higher in the KDLN of the islet/MMV-Gel-transplanted mice. Collectively, the FasL/PD-L1-immobilized MMV-Gel provides an immunosuppressive reservoir to induce immune privilege for allogeneic islet transplantation.

## MMV-Gel fosters survival and function of islet allografts

The therapeutic efficacy of islet/MMV-Gel was evaluated in reversing diabetes on the diabetic mouse model. Blood glucose levels (BGLs) of the diabetic mice were determined after transplantation with 500 islet equivalent (IEQ) of allogeneic islet, islet/L-Gel and islet/MMV-Gel under the left kidney capsule, respectively (Fig. 4a). All the diabetic mice that received different treatments achieved normoglycemia as presented by BGLs lower than 200 mg/dL at 2 d post-transplantation. Median survival time (MST) of the islet allograft was calculated to be 12 d, and all the grafts were eventually rejected at 14 d post-transplantation (Supplementary Fig. 26). L-Gel did not alleviate immunologic rejection of the allogeneic islets. The mice implanted with islet/L-Gel exhibited MST of 12 d and complete rejection within 18 d. As expected, the islet/MMV-Gel-transplanted mice showed significant prolongation of graft survival, 66% of which maintained normoglycemia within 30 d. At 30 d post-transplantation, the kidneys of the mice were harvested and

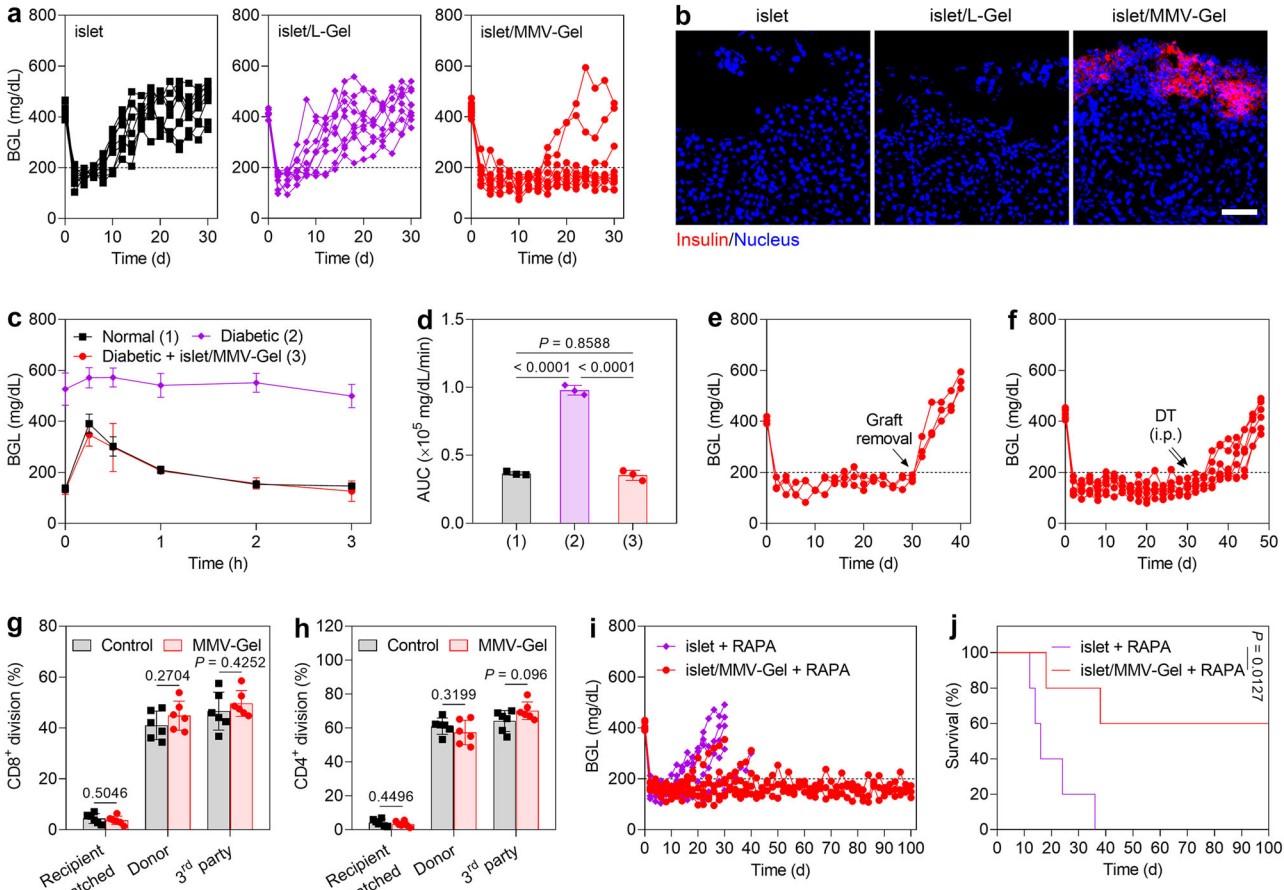

**Fig. 4 | MMV-Gel prolongs survival and function of the transplanted allogeneic islet. a** Changes in the individual BGLs of the diabetic mice after transplantation of islet, islet/L-Gel and islet/MMV-Gel ($n = 9$ mice). **b** Expression of insulin in the transplanted site of the kidney at 30 d post-transplantation of islet, islet/L-Gel and islet/MMV-Gel examined by immunofluorescent staining. Representative is displayed from 3 independent experiments. Scale bar, 50 μm. **c** IPGTT on the normal mice, diabetic mice and diabetic mice at 30 d post-transplantation of islet/MMV-Gel that have survived. **d** AUC values of glucose obtained from IPGTT. Changes in the individual BGLs of the islet/MMV-Gel-transplanted diabetic mice after graft removal (**e**) ($n = 3$ mice) and the islet/MMV-Gel-transplanted FoxP3/DTR diabetic mice after i.p. injection of DT (**f**) ($n = 6$ mice). Black arrows indicate graft

removal or two i.p. injections of DT at Day 30 and 31. Proportion of CD8$^+$ (**g**) and CD4$^+$ (**h**) T cell proliferation in spleen from the mice at 30 d post-transplantation of MMV-Gel to the splenocytes isolated from the BALB/c (recipient-matched), C57BL/6 (donor) and C3H (3rd party) mice determined by flow cytometry. **i** Changes in the individual BGLs of the diabetic mice after transplantation of islet and islet/MMV-Gel with i.p. injection of RAPA. **j** Survival of islet and islet/MMV-Gel combined with RAPA ($n = 5$ mice). Data are shown as mean ± s.d. ($n = 3$ mice in **c**, **d**; $n = 6$ mice in **g**, **h**). One-way ANOVA with Tukey post-hoc test was used for statistical analysis of **d**. Two-tailed unpaired $t$-test was used for statistical analysis of **g**, **h**. Two-sided log-rank (Mantel−Cox) test was used for statistical analysis of **j**. Source data are provided as a source data file.

examined by immunofluorescent staining. Insulin was detected at the transplantation site of the mice treated with islet/MMV-Gel, whereas massive islet destruction with non-insulin-positive architectures were visualized in other treatment groups (Fig. 4b). These results demonstrate that local application of MMV-Gel markedly enhances the survival of islet allograft.

To appraise the metabolic control of blood glucose, intraperitoneal glucose tolerance test (IPGTT)[43] was performed on the islet/MMV-Gel-treated diabetic mice without graft rejection 30 d after implantation. Compared with the diabetic mice that were unable to regulate blood glucose, the mice receiving the islet/MMV-Gel transplantation revealed a similar pattern of blood glucose clearance to the normal mice, as illustrated by BGL returning to normoglycemia within 2 h (Fig. 4c). Area under concentration-time curve (AUC) of glucose during IPGTT was not statistically different between the normal mice and islet/MMV-Gel-treated mice (Fig. 4d). The data indicate that the islet allograft under immuno-protection of MMV-Gel retains glucose-responsive insulin-secretory function equivalent to the normal islet for blood glucose regulation. In addition, surgical removal of MMV-Gel containing the islet (graft removal) caused hyperglycaemia as

evidenced by BGL exceeding 200 mg/dL (Fig. 4e), confirming that the diabetes reversal is dependent on the islet/MMV-Gel transplantation.

To assess that Treg cells were also responsible for enhanced graft survival, two intraperitoneal (i.p.) injections of diphtheria toxin (DT) (50 ng/g) were implemented to the islet/MMV-Gel-implanted FoxP3/DTR diabetic mice to directly deplete the Treg cells[44], followed by monitoring BGL variation. Early treatment with DT at Day 14 and 15 led to rejection of all grafts within the following 18 d (Supplementary Fig. 27). Meanwhile, the islet/MMV-Gel-transplanted diabetic mice receiving the DT treatment at Day 30 and 31 experienced similar graft rejection (Fig. 4f). The data indicate that depletion of Treg cells causes rejection of graft under the established immune acceptance transplantation microenvironment. By comparison, the normal FoxP3/DTR mice treated with DT did not show any noticeable increase of BGL that was in the range of normoglycemia (Supplementary Fig. 28). The results substantiate that Treg cells play an essential role on allograft acceptance for the mice receiving the FasL/PD-L1-presenting MMV-Gel.

To further verify that local implantation of MMV-Gel did not pose a risk of systemic immunosuppression, systemic response of the mice transplanted with MMV-Gel to the donor antigen was evaluated by

mixed lymphocyte reaction (MLR) assay[45]. The BALB/c (recipient-matched) mice were used as negative control, while the C3H (third-party) mice were employed as positive control. Both of CD8[+] and CD4[+] T cells in the spleen of the mice implanted with MMV-Gel generated a strong proliferative response to either donor or third-party stimulators, but not to recipient-matched antigens, which presented a similar magnitude to those of the normal mice (Fig. 4g, h and Supplementary Fig. 29). Moreover, routine blood analysis showed no significant reduction in the counts of PLTs, neutrophil and lymphocyte in the blood of the mice treated with MMV-Gel during the studied time period (Supplementary Fig. 30). Thus, the MMV-Gel-treated mice maintain systemic immune competence, suggesting that the immunosuppressive effect induced by the FasL/PD-L1-engineered MMV-Gel is localized at the transplantation site for protection of the allografts.

The long-term function for graft acceptance was estimated on the diabetic mice by combination of inoculation of islet/MMV-Gel with repeated administration of low dosage of RAPA. RAPA as the first-line non-steroidal immunosuppressive drug was used for reducing the immune pressure on the allograft in the early post-transplantation period. Combination treatment with RAPA notably prolonged the survival of allogeneic islet with MST of 16 d (Fig. 4i, j). In stark contrast, 60% of the mice treated with islet/MMV-Gel under transient cover of RAPA showed sustained survival of graft up to an observation period of 100 d. These results indicate that local implantation of MMV-Gel in combination with systemic treatment with RAPA results in persistent immuno-protection of a significant proportion of allografts for long-term therapeutic efficacy.

Safety evaluation of MMV-Gel was conducted by monitoring levels of serum biomarkers of liver and kidney functions and histologically examining normal tissues at 60 d post-transplantation. The serum levels of the studied biomarkers were not statistically different between the MMV-Gel-treated and untreated mice (Supplementary Fig. 31). Haematoxylin and eosin (H&E)-stained images revealed no obvious pathological changes of the studied normal tissues (Supplementary Fig. 32). The results suggest that the MMV-Gel strategy is acceptably safe within the studied period.

### MMV-Gel prolongs survival of skin allograft

The immuno-protective efficacy of MMV-Gel was assessed on allogeneic skin transplantation that remains a big challenge identified by extremely strong immunological rejection responses[46]. MMV-Gel was applied on the wound of the BALB/c mouse after skin removal, and the harvested donor full-thickness skin from the C57BL/6 mouse was placed on the hydrogel and sutured. After 7 d, the T cell infiltration in the transplanted skin was examined by immunofluorescent staining and analysed by flow cytometry. As expected, the confocal microscopic images showed that accumulation of both CD8[+] and CD4[+] T cells evidently reduced in the skin allograft after the MMV-Gel treatment (Fig. 5a and Supplementary Fig. 33). The flow cytometric quantitative analysis validated that the MMV-Gel-applied mice had significant decrease of Teff cells with increase of Treg cells in the transplanted skin (Fig. 5b–d and Supplementary Fig. 34). Moreover, skin-draining lymph node (SDLN) and spleen were harvested, and the proportion of the infiltrating T cells in these tissues was determined by flow cytometry. Treatment with MMV-Gel also conspicuously reduced the Teff percentage and increased the Treg proportion in SDLN (Fig. 5e–g and Supplementary Fig. 35). No marked elevation of Treg cell proportion was determined, whereas significant decrease of Teff cell accumulation within spleen (Fig. 5h–j and Supplementary Fig. 36). The results afford straightforward in vivo evidence of immunomodulatory microenvironment established by MMV-Gel for allogeneic skin acceptance.

The appearance of skin allograft was further monitored to evaluate the capacity of MMV-Gel to prolong the survival of skin allograft (Fig. 5k). The transplanted skin was reduced, shrivelled and scabbed within 14 d, which is indicative of a strong immune exclusion[47]. The application of L-Gel did not improve the grafting efficacy. In marked contrast, the skin allograft treated with MMV-Gel was nearly intact and well attached to the wound area. The skin survival after different treatments was also monitored by observing complete sclerosis and crusting of the skin allograft (Fig. 5l). The MMV-Gel treatment noticeably extended the survival of the transplanted skin compared with either non-treatment or the L-Gel treatment. Encouragingly, 60% of the skin allograft survived after treatment with MMV-Gel within 14 d. At 14 d post-grafting, the untreated or L-Gel-treated skins were necrotic and desquamated from the recipient skin, while the MMV-Gel-treated skin maintained intact morphology and was well adhered to the recipient skin (Fig. 5k). In addition, SDLNs harvested from the mice after the skin/MMV-Gel treatment were smaller and lighter than that after other treatments (Fig. 5m, n). No statistical difference in the SDLN weight between the skin/MMV-Gel-treated mice and normal mice. Collectively, the FasL/PD-L1-presenting MMV-Gel with immuno-protective effects prolongs the skin allograft survival and promotes tissue repairing.

## Discussion

Allogeneic transplantation requires evasion from the immune system to prevent destruction and rejection of allografts. We have developed an immuno-protective niche using the MSC-derived vesicle-tethered hydrogel, MMV-Gel for acceptance of allograft with long-term function and survival in the absence of systemic immunosuppression. MMV-Gel is readily prepared by mixing the thiol-decorated MMVs and a-HA via a thiol-acrylate Michael addition reaction, which is fabricated efficiently and selectively under mild conditions without any chemical initiators for avoiding function impairment and viability decline of the encapsulated grafts. Upon formulation optimization, MMV-Gel possesses favourable mechanical properties as the entire device is soft but tough to maintain its shape, three-dimensional supporting for stably holding grafts, and interconnected porosity allowing diffusion of molecules including glucose and insulin, which is a promising device for transplants with potential clinical translation. As validated, MMV-Gel with prolonged local retention supports long-acting bioactivity of the FasL/PD-L1-presenting MMVs on inducing the Teff cell apoptosis and boosting the Treg cell generation for construction of local immuno-protective microenvironment.

Transplantation with the allogeneic islets that are embedded in MMV-Gel is effective in reversing diabetes on the diabetic mouse model, significantly prolonging the life span of the islets and maintaining their function for glycaemic control. The mice transplanted with the allogeneic islets exhibited high proportions of CD8[+] and CD4[+] Teff cells at all of the transplant site (kidney), KDLN and spleen. This strong and sustained systemic immune response is due to severe immune rejection at the transplanted site, which contributes to the dysregulation of T cells in secondary lymphoid organs including the elevated frequency of host Teff cells in the spleen. By comparison, islet/MMV-Gel potently suppressed the local immune response, as evidenced by significant reduction of Teff cell proportion within the transplant site, which allows the host to accept the allografts but maintains the normal immune response. In addition, the T cells in the spleens from the normal recipient mice implanted with MMV-Gel showed a proliferative response to the donor and third-party antigens comparable to that from the untreated mice, suggesting that MMV-Gel confers local rather than systemic immunosuppression for allografts. Therefore, MMV-Gel offers localized nature of immune protection without causing systemic unresponsiveness, which averts the risk of opportunistic infections in common with the clinically-used immunosuppressive drugs that induce non-specific immunosuppression. The immuno-protective effect of MMV-Gel is further disclosed in connection with the elevated frequency of Treg cells within the transplant site.

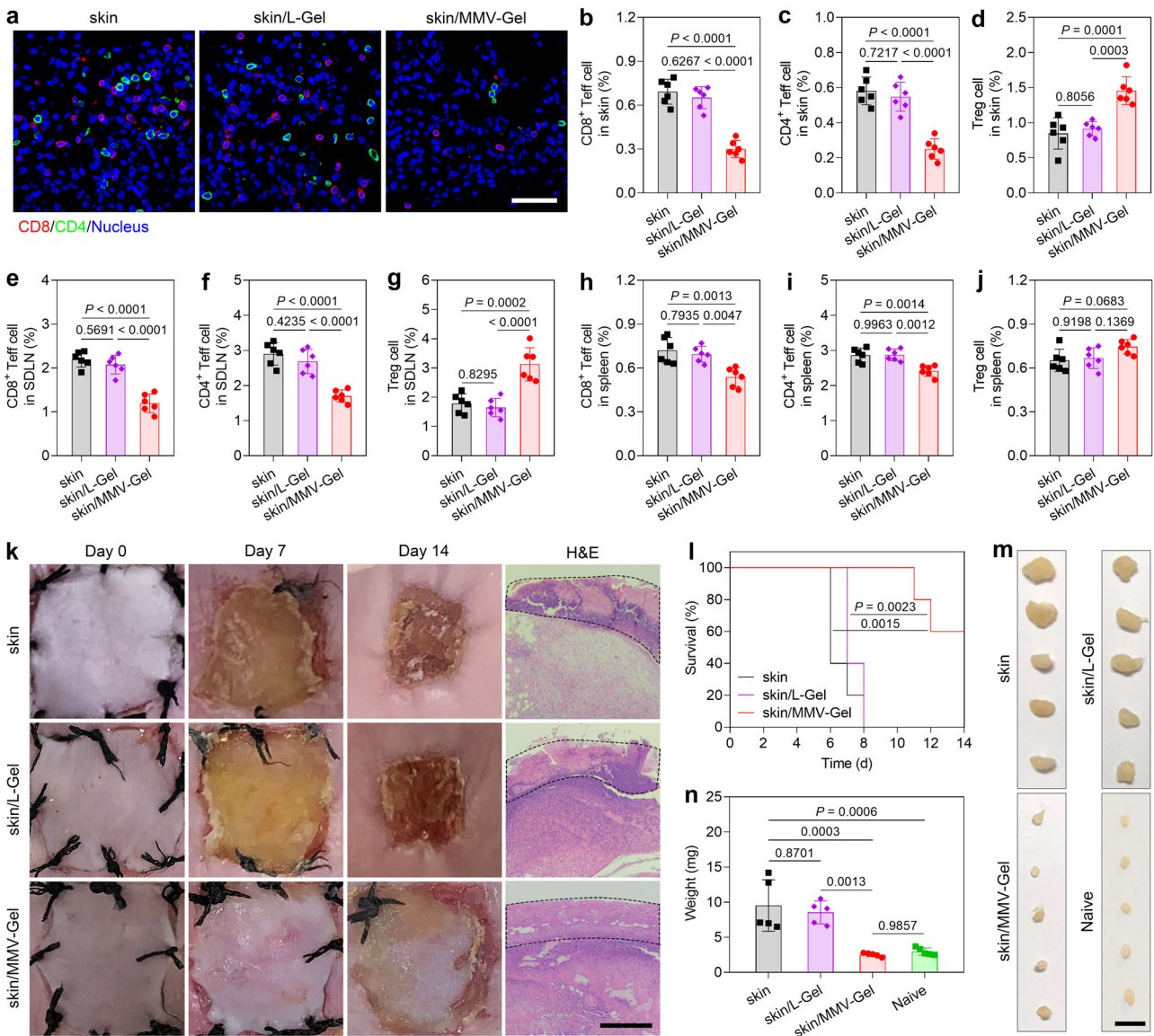

**Fig. 5 | MMV-Gel prolongs survival of the transplanted allogeneic skin.**
**a** Infiltration of CD8[+] and CD4[+] T cells in skin graft at 7 d post-transplantation of skin, skin/L-Gel and skin/MMV-Gel determined by immunofluorescent staining. Representative is displayed from 3 independent experiments. Scale bar, 50 μm. Proportions of CD8[+] Teff (CD8[+]CD44[+]CD62L[−]), CD4[+] Teff (CD4[+]CD44[+]CD62L[−]) and Treg (CD4[+]FoxP3[+]) cells in the skin graft (**b**–**d**), SDLN (**e**–**g**) and spleen (**h**–**j**) at 7 d post-transplantation of skin, skin/L-Gel and skin/MMV-Gel determined by flow cytometry. **k** Images of skin graft at 0, 7 and 14 d post-transplantation of skin/ L-Gel and skin/MMV-Gel. Histological examination of the transplanted area at 14 d post-transplantation by H&E staining. Representative is displayed from 5 independent experiments. Black dotted lines indicate the graft. Scale bar, 400 μm. **l** Survival of skin, skin/L-Gel and skin/MMV-Gel ($n = 5$ mice). Images (**m**) and weights (**n**) of SDLN harvested from the mice at 14 d post-transplantation of skin, skin/L-Gel and skin/MMV-Gel. Scale bar, 5 mm. Data are shown as mean ± s.d. ($n = 6$ mice in **b**–**j**; $n = 5$ mice in **n**). One-way ANOVA with Tukey post-hoc test was used for statistical analysis of **b**–**j**, **n**. Two-sided log-rank (Mantel−Cox) test was used for statistical analysis of **l**. Source data are provided as a source data file.

Treg cells have been demonstrated for maintaining the transplant homeostasis[48]. Co-transplantation of Treg cells verifies prolongation of the islet survival[49]. However, mass production of Treg cells and maintenance of their functions remain challenging. Our strategy using the FasL/PD-L1-engineered MMV-Gel brings about a transplant microenvironment with enrichment of resident Treg cells by specially eliminating the activated alloreactive Teff cells through the Fas/FasL apoptosis and promoting generation of iTreg cells via the PD1/PD-L1 interaction for enhanced allograft acceptance. Encouragingly, a single application of islet/MMV-Gel in combination with a short-term and low-dose administration of RAPA attains sustained prevention of allogeneic transplantation rejection for long-term acceptance of the islet allograft, providing a potential combination therapeutic strategy that maintains effectiveness while minimizing the utilization of

immunosuppressive therapies to prevent graft damage from drug-related toxicity.

We further verify on the clinically-relevant skin allograft mouse model with stronger immune rejection response that MMV-Gel improves the survival of skin allografts in the absence of immunosuppressants, which offers a promising approach for management of the patients requiring large-area skin transplantation commonly associated with systemic infections that disable using immunosuppressive regiments. MMV-Gel reduces the CD8[+] and CD4[+] T cell penetration in the transplanted region, which directly counteracts the rejection caused by immunity unique to high antigen skin transplantation. In addition to weakening the CD8[+] and CD4[+] Teff cells from the graft site, SDLN and spleen, MMV-Gel induces the activated effector phenotypes of Treg cells in a non-redundant way that preferentially

home to the transplanted skin and SDLN rather than distant spleen, suggesting that MMV-Gel elicits potent immunosuppression[50]. Application of MMV-Gel delays acute rejection and provides an immuno-protective microenvironment for the integration of reparative cells to facilitate the survival of allogenic skin. The environmental milieu of SDLN of the mice transplanted with skin/MMV-Gel is less pro-inflammatory, indicating that MMV-Gel attenuates inflammation to suppress alloimmune responses for allogeneic skin acceptance. In addition, the safety assessment shows that MMV-Gel does not cause function impairment and pathological variations of normal tissues.

Considerable efforts have been made to evade the immune attack for allograft transplantation. PD-L1-expression islets are engineered by gene transfection to prevent from acute rejection[51]. This approach is a straightforward but subject to uncontrollable expression of immuno-modulatory proteins. An alternative strategy is encapsulation of islets in the device composed of semipermeable materials to hinder the penetration of immune cells, thereby decreasing the contact of immune cells to the embedded islets[52]. However, it is challenging but also significant to achieve a suitable device that is biocompatible without causing foreign-body reactions and minimises stress on the grafts without impairing their survival and function for immunoisola-tion. Biomaterial scaffolds are gaining increasing attractions for islet transplantation, which possess appropriate pore sizes that can accommodate islets and provide modifiable interfaces for protein immobilisation. Typical examples of modifying PD-L1[53] or FasL[54,55] via controllable biotin-streptavidin coupling support local immunomo-dulation for allogeneic islet transplantation that advances cell-based engraftment, which focuses on a single immunomodulatory protein for suppressing immune rejection. In this study, the biological prop-erties of MMVs are well leveraged for immune modulation, and MMV-Gel is readily fabricated and used for transplants, which avoids the complications associated with graft engineering or protein modifica-tion and allows more flexibility in transplantation. Meanwhile, MMVs are firmly tethered within the hydrogel network to achieve persistent immunomodulatory effect. Apart from presentation of FasL and PD-L1, MMVs harbour a collection of immunomodulatory surface biomole-cules, which enables MMV-Gel to be an immunosuppressive protein compounding depot.

Altogether, we demonstrate that the MMV-Gel-based immuno-protective niche rendering of costimulatory molecules, FasL and PD-L1 achieves immune privilege for allografts. We anticipate that this strategy will offer new opportunities for utilizing MMVs as an immu-nomodulatory element that are engineered in the hydrogel scaffold, which can lead to the development of increasingly immuno-protective device for allotransplantation. Here allogeneic islet and skin were encapsulated in and attached on the hydrogel, respectively; however, other allografts could also be delivered and protected by the immu-noregulatory hydrogel implant, such as small-diameter vascular grafts[56] and pluripotent stem cell-based cellular grafts[42,57]. Moreover, in addition to endogenous MMVs, the compositions used in the for-mulation including HA and DSPE-PEG are biocompatible and approved by FDA, promoting the clinical translation. Additional investigations in large animal models will be necessary for further proof of efficacy and translation to the clinic.

## Methods
### Materials
HA sodium salt (110 kDa) was provided from Freda Biochem. Acrylic anhydride was obtained from Aladdin Bio-Chem Technology. DSPE-PEG-SH was acquired from Ponsure Biological (Shanghai, China). Fluorescence-labelled CD29, CD90.2, CD34, CD45, CD3, CD4, CD8a, CD44, CD62L, FoxP3, PD1, PD-L1, Fas and Nrp-1 antibodies were pro-vided by BioLegend, while fluorescence-labelled FasL antibody was purchased from Invitrogen. FasL and PD-L1 antibodies were purchased from Affinity. HRP-labelled IgG antibody was acquired from

Proteintech. PD-L1 (KOA0875) and MIP-1α (KOA0258) ELISA kits were purchased from Rockland. MCP-1 (1217392) and IL-6 (1210602) ELISA kits were provided by Dakewe. FasL (ZY-FASL-MU) ELISA kit was obtained from Zeye.

### Animal models
The BALB/c and C57BL/6 mice (female, ~20 g) were provided by Comparative Medicine Centre of Yangzhou University. The C3H mice (female, ~20 g) were purchased from Beijing Vital River Laboratory Animal Technology. The Foxp3/DTR mice (female, ~20 g) were obtained from the Shanghai Model Organisms Center, Inc. All the animals were treated according to the Guide for Care and Use of Laboratory Animals, approved by the Animal Experimentation Ethics Committee of China Pharmaceutical University (2021-06-024). All animals were housed under constant temperature and humidity at a standardised facility with free access to water and food under pathogen-free conditions.

### Cell models
For isolation of MSCs, the femurs from the BALB/c mice were collected and flushed with phosphate-buffered saline (PBS). After centrifugation (300 × g, 5 min), the collected cells were cultured in MEM α medium containing penicillin-streptomycin (1×) and fetal bovine serum (FBS) (20%) in the incubator (HERAcell 150i, Thermo Scientific). The fresh medium was supplemented every 3 d. MSCs were characterized by specific surface markers including CD29, CD90, CD34 and CD45, and analysed by flow cytometer (CytoFlex S, Beckman). For upregulation of the PD-L1 expression, MSCs (1 × 10$^6$ cells) were incubated with IFN-γ (1 ng/mL) for 12 h. The FasL and PD-L1 expression on MSCs was ana-lysed by flow cytometry.

For isolation of RBCs and PLTs, heparinized blood (10 mL) col-lected from the BALB/c mice was centrifuged (100 × g, 20 min). After washed with cold PBS twice, the pellet of RBC was used for further experiment. The supernatant plasma (4 mL) was added to the Tris-HCl buffer (4 mL, pH 6.5) containing prostaglandin E1 (1 μM) and EDTA (5 mM). The mixture was centrifuged (800 × g, 20 min) to obtain the resting PLTs. To activate PLTs, the resting PLTs were incubated with thrombin (10 μM) at 25 °C for 10 min. The activation process was ter-minated by adding paraformaldehyde (2%) to obtain aPLTs. The expression of FasL on RBCs, PLTs and aPLTs was analysed by flow cytometry.

For isolation of T cells, the spleen was crushed, homogenized and filtered through a cell strainer (40 μm). The single cell suspension was further incubated with the RBC lysis buffer at 25 °C for 3 min to obtain the splenocytes. The splenocytes were labelled with CD3 fluorescence antibody, and T cells were sorted by flow cytometer (FACSAria II, BD Biosciences).

For isolation of pancreatic islets from the C57BL/6 mice, the pancreas was perfused with collagenase V (4 mL, 1 mg/mL) through the common bile duct, and then excised. After 20 min of digestion at 38 °C, the pancreas was shaken in cold RPMI-1640 medium (40 mL) containing FBS (10%). The digested pancreatic tissues were filtered through the mesh (600 μm) and the cell suspension was washed with cold Hanks balanced salt solution twice by centrifugation (100 × g, 1 min). The precipitate was purified by the Ficoll density gradient (25%, 23%, 20%, 11%) and centrifuged (300 × g, 25 min) to aspirate islets at the interface of the 23% to 20% and 20% to 11%. The obtained islets were cultured in RPMI-1640 medium containing penicillin-streptomycin (1×) and FBS (10%) in the incubator.

### Preparation and characterization of MMVs
MSCs were incubated in the Tris buffer (30 mM) containing EGTA (0.5 mM), sucrose (75 mM), mannitol (225 mM), BSA (0.5%) and protease inhibitors (1×) at 4 °C overnight. The cell suspension was sonicated for 3 min, followed by centrifugation (22,000 × g, 0.5 h). The supernatant

was withdrawn, followed by centrifugation (120,000 × g, 1 h). The pellet of cell membrane was collected, followed by addition of PBS and 3 min of sonication. MMVs were obtained after extrusion through membrane filter (100 nm). MMVs was quantified by BCA protein assay kit (Yeasen Biotech). Protein expression on MMVs were characterized by SDS-PAGE (10%), while the FasL and PD-L1 expression was examined by western blotting and ELISA kits. For evaluation of the PD-L1 retention on the surface of MMVs from the IFN-γ-treated MSCs, the PD-L1 surface expression within 30 d was determined by ELISA kit. MMVs were measured by Zetasizer (Nano ZS90, Malvern) and visualized by TEM (HT7700, Hitachi). For modification of thiol groups, MMVs were incubated with DSPE-PEG-SH at a ratio of 1:3 (mg protein: mg) at 37 °C for 15 min, followed by washing with PBS twice by ultrafiltration tube (10 k MWCO) (Millipore). The thiol-modified MMVs was quantified to 160 mg protein/mL with PBS to obtain the MMV solution.

For apoptosis assay, the splenocytes ($1 \times 10^5$ cells) were treated with Con A (5 μg/mL) for 24 h, and the cells were incubated with MMVs for 48 h. The apoptotic percentage of the gated CD3$^+$ T cells within the splenocytes were determined by Annexin V-FITC/PI apoptosis detection kit (BD Biosciences). For proliferation assay, the Con A-treated splenocytes were labelled with CFSE (5 μM) for 10 min. The labelling process was terminated by adding FBS (10%). After washed with medium twice and additional 72 h of incubation with MMVs, the CFSE-labelled splenocytes were examined by immunofluorescent staining, and then incubated with PI (1 μg/mL). The division of the gated CD3$^+$ T cell within the splenocytes was assessed with a diluted dye. In the blocking experiment, FasL or PD-L1 antibodies (1 μg/mL) were added. For characterization of T cell phenotype, the splenocytes were treated with Con A (5 μg/mL) for 24 h, and the cells were incubated with MMVs (20 μg) for 48 h. The cells were stained with the specific antibodies to determine CD8$^+$ Teff cell (CD8$^+$CD44$^+$CD62L$^-$), CD4$^+$ Teff cell (CD4$^+$CD44$^+$CD62L$^-$), Treg cell (CD4$^+$FoxP3$^+$) and analysed by flow cytometry.

## Synthesis and characterization of a-HA

Deionized water (50 mL) dissolving HA sodium salt (1 g) was added with acrylic anhydride (1.6 mL) under stirring, the pH value of which was held in the range of 8 to 9 for 24 h. a-HA was obtained by acetone precipitation and ethanol wash (× 2), followed by dialysis against deionised water. The purified a-HA was obtained by lyophilisation. The acrylated modification degree was calculated as ~ 17% based on the $^1$H NMR spectrum. $^1$H NMR (D$_2$O, 500 MHz, $\delta$ ppm): 6.44 (s, 1H, CH$_1$H$_2$ = CCO), 6.23 (s, 1H, CH$_1$H$_2$ = CCO), 6.00 (s, 1H, CH$_1$H$_2$ = CCO), 2.01 (s, 3H, NHCOCH$_3$), 1.86–1.97 (m, 3H, CH$_2$ = C(CH$_3$)CO).

## Preparation and characterization of MMV-Gel

The thiol-modified MMV solution (100 μL, 160 mg protein/mL, pH 7.4) was added into the a-HA solution (100 μL, 50 mg/mL, pH 7.4). After 1 h of incubation at 25 °C, MMV-Gel was formed. L-Gel was prepared by mixing a-HA and the thiol-modified liposomes instead of MMVs.

The rheology of MMV-Gel during gelation was monitored by rheometer (MARS60, Thermo Scientific). The morphology of MMV-Gel was observed by SEM (Regulus8100, Hitachi). Rho-MMVs were obtained by incubating MMVs with rhodamine isothiocyanate at 37 °C for 15 min, followed by washing with PBS thrice. Rho-MMV-Gel was visualized by confocal microscope (FV3000, Olympus). To assess the effect of MMV-Gel on inducing the T cell death in vitro, the T cells sorted from the splenocytes were treated with Con A (5 μg/mL) for 24 h, followed by seeded on the surface of MMV-Gel. After 48 h, the viability of the activated T cells infiltrating in the hydrogel was examined by calcein AM/PI double-staining assay and observed by confocal microscope. To evaluate the binding efficiency of PD-L1 on MMV-Gel to PD1 of T cells in vitro, the T cells sorted from the splenocytes were treated with Con A (5 μg/mL) for 24 h, followed by seeded on the surface of MMV-Gel. After 24 h, the expression of PD1 on the activated

T cells infiltrating in the hydrogel was examined by fluorescence-labelled PD1 antibody staining and observed by confocal microscope.

## Function and viability of islet/MMV-Gel

The islets were mixed with the a-HA solution, followed by adding the MMV solution. After 1 h of stabilisation at 25 °C, islet/MMV-Gel was obtained. The morphology of islet/MMV-Gel was observed by cryo-SEM (SU8010, Hitachi). The production of insulin and glucagon in islet/MMV-Gel was monitored by immunofluorescent staining. The function of islet/MMV-Gel was assessed by 3 mM, 11 mM and 3 mM sequential GSIS assay. The stimulation index of insulin was calculated as the insulin secretion value after high glucose stimulation divided by that after the first low glucose stimulation. The viability of islets in MMV-Gel was evaluated by acridine orange/PI double-staining assay and observed by confocal microscope. The secretion of pro-inflammatory cytokines from islet/MMV-Gel was measured by the corresponding ELISA kits.

To evaluate the in vitro immuno-protective capacity of MMV-Gel, the islets were isolated from the donor C57BL/6 mice, while the T cells were sorted from the splenocytes of the recipient BALB/c mice. After 24 h of treatment with Con A (5 μg/mL), the T cells were 48 h of incubation with islet/MMV-Gel. Viability of islets in MMV-Gel was evaluated by acridine orange/PI double-staining assay and observed by confocal microscope. The function of insulin secretion of islet/MMV-Gel was assessed by GSIS assay.

## Evaluation of acceptance of allogeneic islet transplantation

For establishment of the diabetic mouse models, the BALB/c mice received a single i.p. injection of streptozotocin (220 mg/kg). The BGLs were monitored every two days. The mice with two consecutive BGLs higher than 350 mg/dL were identified as diabetic mice. The diabetic BALB/c mice were transplanted with islet, islet/L-Gel or islet/MMV-Gel (islet from the C57BL/6 mouse) under the left kidney capsule at 500 IEQ per mouse. Rejection of islet allograft was defined by two consecutive BGLs above 200 mg/dL, and the corresponding islet survival was determined. At 7 d post-transplantation, the CD8$^+$ and CD4$^+$ T cell infiltration within the transplantation area of the kidney was examined by immunofluorescent staining and visualized by confocal microscope, while the proportion of the Teff and Treg cells in the transplantation site, KDLN and spleen was examined using immunofluorescent staining and analysed by flow cytometry. The cells were stained with the specific antibodies to determine CD8$^+$ Teff cell (CD8$^+$CD44$^+$CD62L$^-$), CD4$^+$ Teff cell (CD4$^+$CD44$^+$CD62L$^-$), Treg cell (CD4$^+$FoxP3$^+$), apoptotic CD8$^+$ Teff cell (CD8$^+$CD44$^+$CD62L$^-$TUNEL$^+$), apoptotic CD4$^+$ Teff cell (CD4$^+$CD44$^+$CD62L$^-$TUNEL$^+$) and iTreg cell (CD4$^+$FoxP3$^+$Nrp-1$^-$). At 30 d post-transplantation, insulin secretion at the transplanted site of the kidney were examined by immunofluorescent staining and observed by confocal microscope. The islet/MMV-Gel-transplanted diabetic mice were surgical removal of islet/MMV-Gel at 30 d post-transplantation, while the islet/MMV-Gel-grafted FoxP3/DTR diabetic mice were treated by two i.p. injections of DT (50 ng/g) at predetermined time intervals. The BGL changes of the mice were monitored.

Glucose tolerance was assessed by IPGTT. The islet/MMV-Gel-transplanted diabetic mice with graft acceptance for 30 d were fasted overnight and treated by i.p. injection of glucose (2 g/kg). The normal mice and untreated diabetic mice were used as control groups. The BGLs were monitored within 3 h after injection.

For the MLR assay, the BALB/c mice were implanted with MMV-Gel under the left kidney capsule. After 30 d, the splenocytes isolated from the mice were labelled with CFSE (5 μM) for 10 min. The labelling process was terminated by adding FBS (10%). After washed with medium twice, the CFSE-labelled splenocytes were incubated with the mitomycin C-treated splenocytes from the BALB/c (recipient-matched), C57BL/6 (donor) and C3H (third-party) mice for 72 h. The cells

were examined by immunofluorescent staining, followed by 10 min of incubation with PI (1 µg/mL) to exclude the dead cells. The division of the gated $CD8^+$ and $CD4^+$ T cells within the live splenocytes was assessed with a diluted dye and plotted as percent division for the cell population.

For evaluation of the long-term therapeutic efficacy of islet/MMV-Gel in combination with RAPA, the diabetic mice were transplanted with islet or islet/MMV-Gel under the left kidney capsule, and treated by i.p. injection of RAPA (0.2 mg/kg) every day for 15 doses. The BGLs were monitored every two days. Rejection of islet allograft was defined by two consecutive BGLs above 200 mg/dL, and the corresponding islet survival was determined.

### Evaluation of acceptance of allogeneic skin transplantation
The recipient BALB/c mouse was anesthetized, fixed and formed the right skin defect area on the back. The full-thickness trunk back skin was harvested from the donor C57BL/6 mouse and placed in saline. MMV-Gel was implanted on the wound of the recipient mouse, and then the donor skin graft was sutured onto the MMV-Gel layer in the transplanted area, followed by secured with a bandage. Rejection was defined as complete sclerosis and scabbed of skin allograft, and the corresponding skin survival was determined. At 7 d post-transplantation, the $CD8^+$ and $CD4^+$ T cell infiltration within the transplanted skin was examined by immunofluorescent staining and observed by confocal microscope, while the proportion of Teff and Treg cells in the skin graft, SDLN and spleen was examined using immunofluorescent staining and analysed by flow cytometry. The images of the transplantation area were captured at varying days after transplantation. At 14 d post-transplantation, the skin at the transplanted site was histologically examined by H&E staining and observed by optical microscope (BX53, Olympus), and the image and weight of SDLN from the mice were recorded.

### Safety evaluation
The BALB/c mice were implanted with MMV-Gel under the left kidney capsule and the contralateral kidney was used as systemic control like other organs. The blood was collected, and the counts of PLTs, neutrophils and lymphocytes were determined by auto haematology analyser (BC-2800 Vet, Mindray). At 60 d post-implantation, the serum levels of hepatorenal function indicators were determined, and the tissues were histologically examined by H&E staining.

### Flow cytometry staining
For surface staining, the cells were incubated with the fluorescence-labelled antibodies for 30 min at 4 °C. For surface and intranuclear dual-staining, the cells were incubated with the fluorescence-labelled antibody for 30 min at 4 °C for surface staining, treated by True-Nuclear™ transcription factor buffer set (BioLegend), and incubated with fluorescence-labelled antibody for 30 min at 4 °C for intranuclear staining. For surface and TUNEL dual-staining, the cells were incubated with the fluorescence-labelled antibody for 30 min at 4 °C for surface staining, treated with paraformaldehyde (4%) for 30 min at 4 °C and ethanol (70%) for 1 h at 4 °C, and stained by Cell Meter™ TUNEL apoptosis assay kit (AAT Bioquest, ABD-22857). All procedures were implemented following manufacturers' instructions.

### Statistical analysis
GraphPad Prism 8.0 software was used for statistical analysis. Statistical tests utilised for each experiment were specified in the legends of figures, and statistical significance was defined as $P < 0.05$.

### Reporting summary
Further information on research design is available in the Nature Portfolio Reporting Summary linked to this article.

## Data availability
The authors declare that all the data supporting the findings of this study are available within the article and supplementary information. Source data are provided with this paper. Source data is available for Figs. 2a–f, 2h, 2l, 2n, 3c–f, 3h-j, 3l–n, 4a, 4c–j, 5b–j, 5l, 5n and Supplementary Figs. 2a-b, 3, 4a-d, 5a, b, 6a, b, 9, 11a-c, 15, 17a, b, 19a–c, 22a, b, 23, 24a–c, 25a–c, 26, 27, 28, 30a–c, 31 in the associated source data file. Source data are provided with this paper.

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

## Acknowledgements

This work was supported by the National Natural Science Foundation of China (81971730, 82273876, 81673381), the Fok Ying-Tong Education Foundation for Young Teachers in the Higher Education Institutions of China (171028), the Project of State Key Laboratory of Advanced Drug Delivery and Release Systems (DSQZ-ZD-200301), the Fundamental Research Fund for the Central Universities (2632022YC02), the Project of State Key Laboratory of Natural Medicines of China Pharmaceutical University (SKLNMZZ202024), and the Project of Jiangsu Key Laboratory of Drug Design and Optimization of China Pharmaceutical University. We acknowledge the Public Platform of State Key Laboratory of Natural Medicines at China Pharmaceutical University for the use of analytical instrumentation facilities, and the Public Platform of Pharmaceutical Animal Experimental Center at China Pharmaceutical University for the use of living imaging system.

## Author contributions

R.M. conceived and supervised the project. Y.W. and R.M. designed the experiments, analysed the data and wrote the paper. Y.W., R.H., Y.L. and M.L. performed the experiments. All the authors discussed the results and commented on the manuscript.

## Competing interests

R.M., Y.W. and R.H. are applying a patent related to this work. The remaining authors declare that there are no competing interests.
