## [Peer Review File · Nature Communications]

Immuno-protective vesicle-crosslinked hydrogel for allogenic transplantationREVIEWER COMMENTS

Reviewer #1 (Remarks to the Author):

This manuscript report on the innovative utilization of MSC membrane vesicles (denoted MMVs) crosslinked with chemically modified hyaluronic acid into an immunomodulatory hydrogels for delivery of allogeneic islets and skin in a mouse model of transplantation. MMC immunomodulation results from surface presentation of MSC-derived FasL and IFN γ -induced PD-L1 which were previously shown to provide localized immunomodulation and prolongation of islet allograft survival. The strengths of this report are the innovative idea which make it suitable for publication in nature communications and the use of multiple multidisciplinary approaches to validate the platform in vitro and in vivo with two models of allotransplantation (islets and skin). This reviewer has noted some weaknesses that needs to be addressed as indicated below. Importantly, the manuscript will need extensive editing to increase the quality of this manuscript.

Introduction:

- refer to patients 'with' type 1 diabetes; revise 'guaranteed dose'; in case of transplantation patients with type 1 diabetes include the risk of recurrence of autoimmunity which adds up to allojection; revise the statement 'superior immunomodulatory properties' since it's not clear what the reference for superior is; revise PDL1 with PD-L1; revise the statement 'increase the phenotype of regulatory T (Treg) cells' since it's not clear how a phenotype can be increased; clarify which cells are involved in the mentioned 'cell-to-cell contact'; revise the term 'allograft tolerance' since tolerance in allografts is not achievable unless a bone marrow transplant is performed; include that one reason MSCs cannot provide long-term immunomodulatory effects is their limited capability to remain localized after transplant

Results:

- in fig.2a and suppl fig.2a indicate whether it's median or mean fluorescent intensity and on the y axis indicate the marker (PD-L1 or FasL). Also, indicate example of flow cytometry data including gating strategy in supplemental data. In fig.2 legend and in results, indicate for how long the IFN γ treatment was performed on MSCs before flow cytometry analysis; clarify how the optimized IFN γ concentration was determined and for how long after treatment PD-L1 remain upregulated in MSCs and MMV after preparation
- Data shown in Fig.2d and supplementary fig.6 and 7 need a control to exclude that effects are due to MMV cytotoxicity; also additional characterization of T cell phenotype after treatment would be beneficial
- Clarify what is the ration indicated in Suppl. Fig.8
- Fig.3c GSIR results should show responses to sequential stimulation with 3mM, 11mM, and then 3mM glucose since dysfunctional islets may have similar profile than what shown in fig.3c,e and the capability to shut down insulin secretion after glucose lowering is critical for functionality
- It would be beneficial to include an image that shows both islets and labeled MMV-Gel and lower magnification larger field images so a larger number of islets in gels can be visualized
- Baseline levels of IL6, MCP1, MIP1a are very low without stimulation with pro-inflammatory cytokine cocktail; it may be helpful to repeat the experiments showed in Suppl Fig.14 in these conditions
- Revise 'acts as a congenial scaffold'

- Clarify how the hydrogel was incubated with the T cells for in vitro studies and how the BALBc T cells were activated and define L-Gel.
- Fig.3i should show standard GSIS results as discussed above; also, the small standard deviations shown in these GSIS results are very unusual for primary islets
- Revise the statement 'the enhanced retention of MMV-Gel also improves the protection and preservation of graft for producing long-term effects' since this conclusion is not supported by the data provided in the section above
- Clarify which kidney (left or right) was implanted and mention that the contralateral kidney was used as systemic control like other organs
- Improve the quality of Fig.3k
- Revise the conclusion that 'The data manifest that MMV-Gel protects the islet allograft against immune rejection by reducing the accumulation of Teff cells by triggering their apoptosis and promoting the generation of Treg cells by reprogramming at the transplantation site.' since graft rejection/survival was not quantified and T cell apoptosis was not determined and Treg induction vs recruitment was not determined
- Fig.3 l,m,n data (%) do not match the data (%) shown in the representative flow panels in suppl. Fig.16; same applies to suppl. Fig.17 and suppl. Fig.18; also, the legend should clarify which populations were classified as Teff and Treg
- Revise 'The ratios of Teff cells were significantly reduced' since % and not ratio were quantified
- Indicate the number of islets (IEQ) transplanted per mouse and revise 'single dose'
- Fig.4a should show blood glucose data at least for 30 days
- Revise 'Intact structure of the islets with insulin secretory function was observed at the transplantation site of the mice treated with islet/MMV-Gel' since neither islet structure nor insulin secretion was tested
- Revise 'were lack of responsiveness'
- The experiments with the anti-CD25 treatment are problematic because (1) this will deplete also activated T cells and (2) anti-CD25 depletion will also cause rejection of any islet transplants so it should be performed earlier also in the control islet transplant groups and it won't allow a definite conclusion that Tregs are responsible for the observed immunomodulatory effect
- Replace 'Self (BALB/c)' with recipient-matched
- Revise 'persistent immunoprotection of allograft for long-term therapeutic efficacy' since 60% survival indicate partial effects and show whether there is statistically significant effects for data shown in Fig.4j
- Replace figure 5a with an image of the skin graft at lower magnification so the graft is visible and the zoomed in image currently show as an insert
- Fig.5k H&E: indicate the graft using dotted line
- In the discussion revise 'to prevent deconstruction'
- Revise 'allowing unobstructed diffusion of oxygen, nutrients and metabolites' since diffusion was not studied
- Revise 'offers localized nature of immune tolerance' since tolerance was not demonstrated
- In the discussion it needs to be discussed why T cell effects were found systemically (spleen) while MLR responses showed only local effects
- The discussion needs to be expanded to include comparisons with similar strategies
- The methods should include preparation of the MMV solution
- The methods should include flow cytometry staining and analysis and indicate which cells were considered Teff and Treg

Reviewer #2 (Remarks to the Author):

Wang et al. in this manuscript report the use of hydrogels crosslinked with membrane vesicles derived from mesenchymal stem cells, referred to as MMV-Gel, as a localized immunomodulatory approach for allogeneic islet and skin transplantation. Mesenchymal stem cells were shown to constitutively express FasL and stimulated with IFN- γ to upregulate PDL1, as two immunomodulatory molecules. Although the concept is quite novel and the results are exciting, there are significant issues with the manuscript as outlined below.

1. The manuscript can significantly be improved by reformatting and correcting various grammatical issues. The introduction section is a mix of results and discussion, and the discussion section is a repetition of results. The introduction should focus on localized immunomodulatory approaches for transplantation to put the study in a broader context, while the discussion section should provide a critical analysis and interpretation of the results within the framework of existing literature.
2. Various studies using PD-L1 and FasL for localized immunomodulation were not cited and discussed (PMID: 32814902; PMID: 36533683; PMID: 35559682; PMID: 35897156 PMID: 32923626; PMID: 32253240; PMID: 3659935 PMID: 31850658)
3. All flow cytometry data should include a gating strategy and the raw data in the form of dot plots should be presented either in the main figures or in the supplement. Without this information, it is difficult to appreciate the results.
4. Instead of plotting data from only n=3 samples and stating, 'representative of 2 or 3 experiments,' data pooled from independent studies (Fig. 2-5) should be presented to increase the sample numbers.
5. There are various inconsistencies in the main text and figure legends or materials and methods. For example, the result section refers to the quantification of FasL and PDL1 in MMV using a quantitative ELISA, while the supp figure 4b legend cites flow cytometry. Similarly, the main text in the results section mentions sorted T cells stimulated with ConA, while the supplement cites splenocytes being stimulated with ConA.
6. Fig. 2d. Apoptotic ratio should be defined, and the data should be presented in the form of dot plots.
7. Fig. 2f. A blocking anti-PDL1 Ab results in higher levels of T cell proliferation as compared with the culture without Ab. What is the contribution of FasL to the inhibition of proliferation? Also, the proliferation of cells without MMV should be shown as a positive control. Again, dot plots should be shown alongside bar graphs. Also, instead of the apoptotic ratio, the percent apoptosis of CD3 T cells over the background, i.e. MMV should be shown. Moreover, MMV from a control cell line without FasL or a blocking anti-FasL Ab should be used to assess the relative contribution of FasL and PDL1 to the observed T cell inhibition.
8. Given the low percentage of the indicated cell populations (\sim 0.2%), the gating strategy and dot plots for Fig. 3l-n should be provided in the supplement. Additionally, there is an inconsistency between Fig. 3l-n and supplementary Fig. 16 that shows a much higher T cell %, 18 for CD8+CD44-62Llow.
9. Fig. 4e, f shows islet grafts with MMV-gel survive over 30 days and survival is maintained by Treg cells. Why were the animals only monitored for \sim 30 days given the excellent survival of the graft? Also, the injection of anti-C25 Ab to deplete Treg cells appears to cause immediate rejection, which is quite unusual per published literature in this model. One would expect a delayed rejection, >10 days after Ab injection. This point should be discussed.
10. Long-term survival data was shown for the cohort transplanted with MMV-gel and rapamycin (Fig. 4i-

j). Rapamycin was introduced without a rationale. If MMV-gel is sufficient on its own to prevent islet graft rejection, why use rapamycin? Also, Data with individual agents, i.e. MMV without IFN- γ treatment should be shown for comparison.

11. Islet transplantation was performed in C57BL/6-to-BALB/c, which is a low responder, and not all controls reject the graft per published literature (PMID: 19364073). Testing the concept in BALB/c-to-C57BL/6, a high responder, will reinforce the drawn conclusion, but not required.

Responses to reviewers' comments

We deeply thank the reviewers for their comments and suggestions. Our detailed responses to the comments have been provided below and the manuscript has accordingly been revised.

Reviewer #1:

This manuscript report on the innovative utilization of MSC membrane vesicles (denoted MMVs) crosslinked with chemically modified hyaluronic acid into an immunomodulatory hydrogels for delivery of allogeneic islets and skin in a mouse model of transplantation. MMC immunomodulation results from surface presentation of MSC-derived FasL and IFN γ -induced PD-L1 which were previously shown to provide localized immunomodulation and prolongation of islet allograft survival. The strengths of this report are the innovative idea which make it suitable for publication in nature communications and the use of multiple multidisciplinary approaches to validate the platform in vitro and in vivo with two models of allotransplantation (islets and skin). This reviewer has noted some weaknesses that needs to be addressed as indicated below. Importantly, the manuscript will need extensive editing to increase the quality of this manuscript.

Introduction:

- refer to patients 'with' type 1 diabetes; revise 'guaranteed dose'; in case of transplantation patients with type 1 diabetes include the risk of recurrence of autoimmunity which adds up to allorejection; revise the statement 'superior immunomodulatory properties' since it's not clear what the reference for superior is; revise PDL1 with PD-L1; revise the statement 'increase the phenotype of regulatory T (Treg) cells' since it's not clear how a phenotype can be increased; clarify which cells are involved in the mentioned 'cell-to-cell contact'; revise the term 'allograft tolerance' since tolerance in allografts is not achievable unless a bone marrow transplant is performed; include that one reason MSCs cannot provide long-term immunomodulatory effects is their limited capability to remain localized after transplant

Response: Thanks so much for the reviewer's positive evaluation and valuable comments. As the reviewer suggested, we have revised the "Introduction" part point-by-point.

Results:

in fig.2a and suppl fig.2a indicate whether it's median or mean fluorescent intensity and on the y axis indicate the marker (PD-L1 or FasL). Also, indicate example of flow cytometry data including gating strategy in supplemental data. In fig.2 legend and in results, indicate for how long the IFN γ treatment was performed on MSCs before flow cytometry analysis; clarify how the optimized IFN γ concentration was determined and for how long after treatment PD-L1 remain upregulated in MSCs and MMV after preparation

Response: We have indicated the mean fluorescent intensity (MFI) and the marker FasL or PD-L1 on the Y axis of Fig. 2a,b and Supplementary Fig. 2a.

The flow cytometry data and gating strategy for Fig. 2a,b are depicted in Supplementary Figs 36 and 37, respectively.

MSCs were treated with IFN- γ for 12 h, which has been added in the results and the figure legend of Fig. 2.

The optimal concentration of IFN- γ to treat MSCs has already been investigated. The data has been shown in Supplementary Fig. 2a. The expression of PD-L1 increased at an IFN- γ -concentration-dependent manner. When the concentration of IFN- γ was higher than 1 ng/mL, the PD-L1 expression reached a plateau. No significant difference in MFI between 1 ng/mL treated group and 5 ng/mL or 10 ng/mL treated group. Thus, the IFN- γ concentration was optimized to be 1 ng/mL.

To clarify how long PD-L1 remains upregulated on MSCs and MMVs after preparation, we monitored the expression of PD-L1 over time. For evaluation of the retention of upregulated PD-L1 on the surface of MSCs after the IFN- γ treatment, the cells were incubated with IFN- γ (1 ng/mL) for 12 h, and after removal of IFN- γ , the expression of PD-L1 on MSCs was quantified by flow cytometry within 7 d. For assessment of the maintenance of PD-L1 on MMVs derived from the IFN- γ -treated MSCs, the expression of PD-L1 on MMVs after preparation was examined using an ELISA kit within 30 d. The results have been shown as Response Fig. 1. The PD-L1 expression on the IFN- γ -treated MSCs retained for 3 d but sharply reduced with the following 4 d, indicating that MSCs cannot preserve elevated exposure of PD-L1 for a long period after removal of IFN- γ (Response Fig. 1a). In stark contrast, PD-L1 persisted on MMVs for 30 d at least (Response Fig. 1b).

Response Figure 1. a, Relative expression of PD-L1 on MSCs within 7 d after removal of IFN- γ determined by flow cytometry. MFI, mean fluorescent intensity. **b**, Relative expression of PD-L1 on the IFN- γ -treated MMVs within 30 d quantified by ELISA assay. Data are shown as mean \pm s.d. ($n = 6$ independent samples).

Data shown in Fig.2d and supplementary fig.6 and 7 need a control to exclude that effects are due to MMV cytotoxicity; also additional characterization of T cell phenotype after treatment would be beneficial

Response: Thanks for the reviewer's pertinent comments. We evaluated the cytotoxicity of MMVs on the murine normal fibroblast (L929) cells as a control by monitoring the apoptosis-inducing and anti-proliferative effects. For apoptosis assay, the L929 cells were incubated with MMVs (20 μ g) for 48 h, and the apoptotic percentage was determined by Annexin V-FITC/PI double-staining assay. MMVs did not induce any significant apoptosis of L929 cells (Response Fig. 2). For anti-proliferation assay, the L929 cells were incubated with MMVs (20 μ g) for 72 h, and the cell viability was determined by CCK-8 assay. The viability of the MMV-treated L929 cells was not significantly different from that of the untreated counterparts (Response Fig. 3). Together, MMVs have no evident effect on inducing apoptosis or attenuating proliferative response of normal cells at the studied amount and period. In addition, we evaluated the apoptosis-inducing and anti-proliferative effects of erythrocyte membrane-derived vesicles (EMVs) as a control on the T cells within the activated splenocytes. EMVs cannot induce apoptosis or inhibit proliferation of T cells within the activated splenocytes (Response Figs 4,5). These results suggest that the effects of MMVs on triggering apoptosis and suppressing proliferation of the activated T cells are ascribed mainly to the upregulated surface expression of bioactive FasL and PD-L1 rather than their cytotoxicity.

For further characterization of T cell phenotype after treatment, the concanavalin A (Con A)-treated splenocytes were incubated with MMVs (20 μg) for 48 h, and the Teff and Treg cells were analysed by flow cytometry. After treatment with MMVs, the proportions of both CD8⁺ Teff (CD8⁺CD44⁺CD62L⁻) and CD4⁺ Teff (CD4⁺CD44⁺CD62L⁻) cells in the activated splenocytes significantly reduced (Response Fig. 6a,b), while the percentage of Treg (CD4⁺FoxP3⁺) cells markedly increased (Response Fig. 6c).

Response Figure 2. **a**, Apoptotic percentages of L929 cells after treatment with MMVs determined by Annexin V-FITC/PI double-staining assay. Representative is displayed from 6 independent experiments. **b**, Total apoptotic percentages of L929 cells after treatment with MMVs determined by Annexin V-FITC/PI double-staining assay. Data are shown as mean \pm s.d. ($n = 6$ independent samples in **b**). $P > 0.05$ (no significance, n.s.) (two-tailed unpaired t -test for **b**). **c**, Gating strategy for flow cytometric analysis on the apoptosis of L929 cells.

Response Figure 3. Viabilities of L929 cells after treatment with MMVs determined by CCK-8 assay. Data are shown as mean \pm s.d. ($n = 6$ independent samples). $P > 0.05$ (n.s.) (two-tailed unpaired t -test).

Response Figure 4. a, Apoptotic percentages of T cells within the activated splenocytes after treatment with EMVs determined by Annexin V-FITC/PI double-staining assay. Representative is displayed from 6 independent experiments. **b,** Total apoptotic percentages of T cells within the activated splenocytes after treatment with EMVs determined by Annexin V-FITC/PI double-staining assay. Data are shown as mean \pm s.d. ($n = 6$ independent samples in **b**). $P > 0.05$ (n.s.) (two-tailed unpaired t -test for **b**). **c,** Gating strategy for flow cytometric analysis on the apoptosis of T cells within the activated splenocytes.

Response Figure 5. **a**, Proliferation of T cells within the activated splenocytes after treatment with EMVs determined by CFSE dilution assay. Representative is displayed from 6 independent experiments. **b**, Proportion of T cells within the activated splenocytes after treatment with RMVs determined by CFSE dilution assay. **c**, Gating strategy for flow cytometric analysis on the proliferation of T cells within the activated splenocytes after treatment with EMVs. Data are shown as mean \pm s.d. ($n = 6$ independent samples in **b**). $P > 0.05$ (n.s.) (two-tailed unpaired t -test for **b**).

Response Figure 6. Proportion of CD8⁺ Teff (a), CD4⁺ Teff (b) and Treg (c) cells in the activated splenocytes after treatment with MMVs determined by flow cytometry. The flow cytometric plots present the cell population percentages in CD8⁺ T cells (a) and CD4⁺ T cells (b,c). Representative is displayed from 6 independent experiments. Data are shown as mean \pm s.d. ($n = 6$ independent samples). ** $P < 0.01$ (two-tailed unpaired t -test).

Clarify what is the ration indicated in Suppl. Fig.8

Response: The hydrogel is formed by the thiol-acrylate Michael addition reaction between the acylate groups of *a*-HA and the modified thiol groups of MMVs. The degree of thiol modification of MMVs is a key factor affecting the gel formation when mixed with *a*-HA, which was altered by different incubation ratios of MMVs and DSPE-PEG-SH (mg protein:mg). The gelation results of the mixture of *a*-HA and the DSPE-PEG-SH-anchored MMVs were monitored by the tube inversion assay to determine the optimal ratio of MMVs and DSPE-PEG-SH, which has been shown as Supplementary Fig. 10 (original Supplementary Fig. 8). The mixture of MMVs (100 μ L, 160 mg protein/mL) and *a*-HA (100 μ L, 50 mg/mL) presented

flowing liquid at the ratios of 1:0, 1:0.15, 1:0.3, and 1:1 of MMV to DSPE-PEG-SH (mg protein:mg), demonstrating that the hydrogel fails to form at these ratios. Contrastingly, the hydrogel formed when the ratios were 1:3 and 1:6. Accordingly, the optimized ratio of MMVs to DSPE-PEG-SH was selected at about 1:3 (mg protein:mg). In addition, the hydrogel could not form when *a*-HA was mixed with the equivalent amount of DSPE-PEG-SH.

Fig.3c GSIR results should show responses to sequential stimulation with 3mM, 11mM, and then 3mM glucose since dysfunctional islets may have similar profile than what shown in fig.3c,e and the capability to shut down insulin secretion after glucose lowering is critical for functionality

Response: As the reviewer suggested, the function of islet/MMV-Gel to secrete insulin in response to glucose change was assessed by 3 mM, 11 mM and 3 mM sequential glucose-stimulated insulin secretion (GSIS) assay. These results illustrated that the islets embedded in MMV-Gel maintained insulin secretion by responding to variation of environmental glucose levels (Fig. 3c). The corresponding calculated insulin stimulation index showed no statistical difference between islet/MMV-Gel and free islet (Fig. 3d). The glucose-responsive insulin secretion capacity of islet/MMV-Gel and free islet was further estimated within 7 d. No significant change of insulin secretion from islet/MMV-Gel or free islet within 3 d, while the decline in the insulin secretion after 7 d was ascribed to the natural attenuation of the islet function under the condition of ex vivo culture (Fig. 3e,f and Supplementary Fig. 15).

Revised Figure 3. c,d, GSIS (**c**) and stimulation index (**d**) of free islet and islet/MMV-Gel. **e,f,** GSIS (**e**) and stimulation index (**f**) of islet/MMV-Gel within 7 d. Data are shown as mean \pm s.d. ($n = 6$ independent samples). $P > 0.05$ (n.s.), $*P < 0.05$ (two-tailed unpaired *t*-test for **d**; one-way ANOVA with Tukey post-hoc test for **f**).

Revised Supplementary Figure 15. GSIS (a) and stimulation index (b) of free islet within 7 d. Data are shown as mean \pm s.d. ($n = 6$ independent samples). $P > 0.05$ (n.s.), $**P < 0.01$ (one-way ANOVA with Tukey post-hoc test for b).

It would be beneficial to include an image that shows both islets and labeled MMV-Gel and lower magnification larger field images so a larger number of islets in gels can be visualized

Response: Thanks for the reviewer's suggestion. The α -HA solution containing islets was incubated with Rho-labelled MMVs (Rho-MMVs) to obtain the islet-embedded Rho-MMV-Gel (islet/Rho-MMV-Gel). The islet was stained with acridine orange. The three-dimensional scanning confocal microscopic images of islet/Rho-MMV-Gel at low and high magnifications have been supplemented in the revision and shown as Supplementary Fig. 14, which showed that the islets (green fluorescence) were embedded within the hydrogel and MMVs (red fluorescence) were distributed around the islets.

Revised Supplementary Figure 14. Three-dimensional reconstructed confocal microscopic images of islet/Rho-MMV-Gel stained with acridine orange at low (a) and high (b) magnification. Representative is displayed from 3 independent experiments. Scale bars, 200 μm (a) and 50 μm (b).

Baseline levels of IL6, MCP1, MIP1a are very low without stimulation with pro-inflammatory cytokine cocktail; it may be helpful to repeat the experiments showed in Suppl Fig.14 in these

conditions

Response: We thank the reviewer for this comment. We have repeated the experiments under these conditions and quantified the pro-inflammatory cytokine expression of each islet instead of the sample volume (Response Fig. 7). The secretion of pro-inflammatory cytokines including MIP-1 α , MCP-1 and IL-6 was not significantly different between the islet/MMV-Gel and free islet groups within one week of culture. We have calculated them as relative values shown in Supplementary Fig. 17.

Response Figure 7. Secretion of MIP-1 α (a), MCP-1 (b) and IL-6 (c) from free islet and islet/MMV-Gel within 7 d. Data are shown as mean \pm s.d. ($n = 6$ independent samples). $P > 0.05$ (n.s.) (two-tailed unpaired t -test).

Revised Supplementary Figure 17. Secretion of MIP-1 α (a), MCP-1 (b) and IL-6 (c) from

islet/MMV-Gel relative to free islet within 7 d. Data are shown as mean \pm s.d. ($n = 6$ independent samples). $P > 0.05$ (n.s.) (two-tailed unpaired t -test).

Revise ‘acts as a congenial scaffold’

Response: We have revised “acts as a congenial scaffold” to “acts as a local depot” in the manuscript.

Clarify how the hydrogel was incubated with the T cells for *in vitro* studies and how the BALBc T cells were activated and define L-Gel.

Response: The islets were isolated from the donor C57BL/6 mice, while the T cells were sorted from the splenocytes of the recipient BALB/c mice. The T cells were treated with Con A (5 $\mu\text{g}/\text{mL}$) for 24 h for activation, and then incubated with islet/MMV-Gel. As shown in Response Fig. 8, islet/MMV-Gel was placed in a confocal dish, and added with the activated T cells, followed by incubation for 48 h. The control gel, L-Gel was prepared by mixing α -HA and thiol-modified liposomes instead of MMVs, which has been defined in the revision.

Response Figure 8. Schematic to show that islet/MMV-Gel is incubated with the activated T cells.

Fig.3i should show standard GSIS results as discussed above; also, the small standard deviations shown in these GSIS results are very unusual for primary islets

Response: We have performed the 3 mM, 11 mM and 3 mM sequential GSIS assay to evaluate the insulin-secreting function of islet/L-Gel and islet/MMV-Gel after 48 h of incubation with the activated T cells. The results with increased sample numbers have been added in Fig. 3i,j. After treatment with the activated T cells, the insulin secretion capacity of islet/MMV-Gel in response to the change of the glucose level retained (Fig. 3i), compared with that of the

untreated islet/MMV-Gel (Fig. 3c), while that of islet/L-Gel was subject to dysfunction. The stimulation index of islet/MMV-Gel was calculated to be significantly higher than that of islet/L-Gel (Fig. 3j). The data indicate that the islets preserve the glucose-responsive insulin-secretory function under protection of MMV-Gel against the T cell-mediated immune attack.

Revised Figure 3. i,j, GSIS (i) and stimulation index (j) of islet/L-Gel and islet/MMV-Gel after incubation with the activated T cells for 48 h. Data are shown as mean \pm s.d. ($n = 6$ independent samples). $P > 0.05$ (n.s.), $*P < 0.05$, $**P < 0.01$, $***P < 0.001$ (two-tailed unpaired t -test).

Revise the statement ‘the enhanced retention of MMV-Gel also improves the protection and preservation of graft for producing long-term effects’ since this conclusion is not supported by the data provided in the section above

Response: This statement has been removed as the reviewer suggested.

Clarify which kidney (left or right) was implanted and mention that the contralateral kidney was used as systemic control like other organs

Response: We have modified the experimental details in the “Methods” part. The results in the revised manuscript to clarify that the implantation site was at the left kidney and the contralateral kidney was used as systemic control like other organs.

Improve the quality of Fig.3k

Response: The quality of Fig.3k has been improved according to the reviewer’s suggestion.

Revised Figure 3k. Infiltration of CD8⁺ and CD4⁺ T cells in the transplanted site of the kidney at 7 d post transplantation of islet, islet/L-Gel and islet/MMV-Gel examined by immunofluorescent staining. Representative is displayed from 3 independent experiments. White dotted lines indicate the graft. Scale bar, 50 μ m.

Revise the conclusion that ‘The data manifest that MMV-Gel protects the islet allograft against immune rejection by reducing the accumulation of Teff cells by triggering their apoptosis and promoting the generation of Treg cells by reprogramming at the transplantation site.’ since graft rejection/survival was not quantified and T cell apoptosis was not determined and Treg induction vs recruitment was not determined

Response: We appreciate the reviewer for the constructive comments. We have quantified graft survival and analysed the proportion of apoptotic Teff and induced Treg (iTreg) cells in the kidney at 7 d post transplantation by flow cytometry. The proportion of apoptotic Teff and iTreg cells was significantly higher in the kidney that received the islet/MMV-Gel treatment (Supplementary Figs 20,21). Median survival time (MST) of the islet allograft in the mice was calculated to be 12 d (Supplementary Fig. 24), and all the grafts were eventually rejected at 14 d post transplantation. L-Gel did not alleviate immunologic rejection of the allogeneic islets. The mice implanted with islet/L-Gel exhibited MST of 12 d and complete rejection after 18 d. Encouragingly, the survival of grafts was significantly prolonged in the mice with the islet/MMV-Gel transplantation and 66% of the islet/MMV-Gel-treated mice maintained normoglycemia within 30 d.

Revised Supplementary Figure 20. Percentages of TUNEL⁺ cell in the CD8⁺ Teff (a) and CD4⁺ Teff (b) cells in the transplanted kidney at 7 d post transplantation of islet, islet/L-Gel and islet/MMV-Gel determined by flow cytometry. Representative is displayed from 6 independent experiments. Data are shown as mean \pm s.d. ($n = 6$ mice). $P > 0.05$ (n.s.), $***P < 0.001$ (one-way ANOVA with Tukey post-hoc test).

Revised Supplementary Figure 21. Percentage of iTreg (CD4⁺FoxP3⁺Nrp-1⁻) cell in Treg cell in the kidney at 7 d post transplantation of islet, islet/L-Gel and islet/MMV-Gel determined by flow cytometry. Representative is displayed from 6 independent experiments. Data are shown as mean \pm s.d. ($n = 6$ mice). $P > 0.05$ (n.s.), $***P < 0.001$ (one-way ANOVA with Tukey post-hoc test).

Revised Supplementary Figure 24. Survival of islet, islet/L-Gel and islet/MMV-Gel ($n = 9$ mice). *** $P < 0.001$ (two-sided log-rank (Mantel–Cox) test).

Fig.3 l,m,n data (%) do not match the data (%) shown in the representative flow panels in suppl. Fig.16; same applies to suppl. Fig.17 and suppl. Fig.18; also, the legend should clarify which populations were classified as Teff and Treg

Response: The quantitative data (%) of Fig. 31-n and Supplementary Figs 19,22,23 (original Supplementary Figs 16-18) show the percentage of Teff or Treg cells in the whole kidney, DLN or spleen, while the flow cytometric analysis plots of Supplementary Figs 19,22,23 (original Supplementary Figs 16-18) present the cell population percentages in the CD8⁺ T or CD4⁺ T cells. We have added the relative descriptions in the legends of these figures.

Revise ‘The ratios of Teff cells were significantly reduced’ since % and not ratio were quantified

Response: We have revised the “ratios” to the “percentages” as the reviewer suggested.

Indicate the number of islets (IEQ) transplanted per mouse and revise ‘single dose’

Response: According to the reviewer’s suggestions, we have clarified that a total of 500 islet equivalent (IEQ) per mouse were transplanted, and modified as “Blood glucose levels (BGLs) of the diabetic mice were determined after transplantation with 500 islet equivalent (IEQ) of allogeneic islet, islet/L-Gel and islet/MMV-Gel under the left kidney capsule, respectively”.

Fig.4a should show blood glucose data at least for 30 days

Response: Changes of BGLs within 30 d have been added in Fig. 4a as the reviewer suggested.

Revised Figure 4a. Changes in the individual BGLs of the diabetic mice after transplantation of islet, islet/L-Gel and islet/MMV-Gel ($n = 9$ mice).

Revise ‘Intact structure of the islets with insulin secretory function was observed at the transplantation site of the mice treated with islet/MMV-Gel’ since neither islet structure nor insulin secretion was tested

Response: We have revised this description according to the reviewer’s comment.

Revise ‘were lack of responsiveness’

Response: We have revised this description as the reviewer suggested.

The experiments with the anti-CD25 treatment are problematic because (1) this will deplete also activated T cells and (2) anti-CD25 depletion will also cause rejection of any islet transplants so it should be performed earlier also in the control islet transplant groups and it won’t allow a definite conclusion that Tregs are responsible for the observed immunomodulatory effect

Response: We appreciate the reviewer’s valuable suggestions. To avoid the possibility that injection of aCD25 for depleting Treg cells might interfere with the Teff cell function, we have used the FoxP3/DTR mice that express the human diphtheria toxin receptor (DTR) in the Treg cells. To assess that Treg cells were also responsible for enhanced graft survival, the islet/MMV-Gel-implanted FoxP3/DTR diabetic mice were treated by two i.p. injections of diphtheria toxin (DT) (50 ng/g) to directly deplete the Treg cells, followed by monitoring the BGL variation. Early treatment with DT at Day 14 and 15 led to rejection of all grafts within the following 18 d (Supplementary Fig. 25). The islet/MMV-Gel-transplanted diabetic mice receiving the DT

treatment at Day 30 and 31 also experienced similar graft rejection, indicating that depletion of Treg cells causes rejection of graft under the established immune acceptance transplantation microenvironment (Fig. 4f). By comparison, the normal FoxP3/DTR mice treated with DT did not show any noticeable increase of BGLs that was in the range of normoglycemia (Supplementary Fig. 26). The results substantiate that the Treg cells play an essential role on allograft acceptance for the mice receiving the FasL/PD-L1-presenting MMV-Gel.

Revised Supplementary Figure 25. Changes in the individual BGLs of the islet/MMV-Gel-transplanted FoxP3/DTR diabetic mice after i.p. injection of DT ($n = 6$ mice). Black arrows indicate two i.p. injections of DT at Day 14 and 15.

Revised Figure 4f. Changes in the individual BGLs of the islet/MMV-Gel-transplanted FoxP3/DTR diabetic mice after i.p. injection of DT ($n = 6$ mice). Black arrows indicate two i.p. injections of DT at Day 30 and 31.

Revised Supplementary Figure 26. Changes in the individual BGLs of the FoxP3/DTR mice after i.p. injection of DT ($n = 6$ mice). Black arrows indicate two i.p. injections of DT at Day 0 and 1.

Replace ‘Self (BALB/c)’ with recipient-matched

Response: We have revised the description in the figures, figure legends and text.

Revise ‘persistent immunoprotection of allograft for long-term therapeutic efficacy’ since 60% survival indicate partial effects and show whether there is statistically significant effects for data shown in Fig.4j

Response: We have revised the description as the reviewer suggested. The statistical significance has been added in Fig. 4j.

Revised Fig. 4j. Survival of islet and islet/MMV-Gel with intraperitoneal injection of RAPA ($n = 5$ mice). $*P < 0.05$ (two-sided log-rank (Mantel–Cox) test).

Replace figure 5a with an image of the skin graft at lower magnification so the graft is visible and the zoomed in image currently show as an insert

Response: We have supplemented the confocal microscopic images of the skin graft at low magnification, which has been shown as Supplementary Fig. 31.

Revised Supplementary Figure 31. Infiltration of CD8⁺ and CD4⁺ T cells in the transplanted skin at 7 d post transplantation of skin, skin/L-Gel and skin/MMV-Gel determined by immunofluorescent staining. Representative is displayed from 3 independent experiments. Scale bar, 200 μ m.

Fig.5k H&E: indicate the graft using dotted line

Response: We have marked the skin graft with the dotted line in Fig.5k as the reviewer suggested.

Revised Figure 5k. Images of the transplanted skin at 0, 7 and 14 d post transplantation of skin, skin/L-Gel and skin/MMV-Gel. Histological examination of the transplanted area at 14 d post transplantation by H&E staining. Representative is displayed from 5 independent experiments. Black dotted lines indicate the grafts. Scale bar, 400 μ m.

In the discussion revise ‘to prevent deconstruction’

Response: We have revised “to prevent deconstruction” to “to prevent destruction”.

Revise ‘allowing unobstructed diffusion of oxygen, nutrients and metabolites’ since diffusion was not studied

Response: We have made a revision revised “allowing unobstructed diffusion of oxygen, nutrients and metabolites” to “allowing diffusion of molecules including glucose and insulin”.

Revise ‘offers localized nature of immune tolerance’ since tolerance was not demonstrated

Response: We have revised “offers localized nature of immune tolerance” to “offers localized nature of immune protection”.

In the discussion it needs to be discussed why T cell effects were found systemically (spleen) while MLR responses showed only local effects

Response: We have supplemented more discussions in the “Discussion” part of the revised manuscript.

The discussion needs to be expanded to include comparisons with similar strategies

Response: We have expanded the discussion to include a comparison with other localized immunosuppression approaches for allotransplantation.

The methods should include preparation of the MMV solution

Response: The preparation of the MMV solution has been supplemented in the “Methods” part of the Supplementary Information.

The methods should include flow cytometry staining and analysis and indicate which cells were considered Teff and Treg

Response: We have added “flow cytometry staining” in the “Methods” part and defined Teff and Treg cells in the revision.

Reviewer #2:

Wang et al. in this manuscript report the use of hydrogels crosslinked with membrane vesicles derived from mesenchymal stem cells, referred to as MMV-Gel, as a localized immunomodulatory approach for allogeneic islet and skin transplantation. Mesenchymal stem cells were shown to constitutively express FasL and stimulated with IFN- γ to upregulate PDL1, as two immunomodulatory molecules. Although the concept is quite novel and the results are exciting, there are significant issues with the manuscript as outlined below.

1. The manuscript can significantly be improved by reformatting and correcting various grammatical issues. The introduction section is a mix of results and discussion, and the discussion section is a repetition of results. The introduction should focus on localized immunomodulatory approaches for transplantation to put the study in a broader context, while the discussion section should provide a critical analysis and interpretation of the results within the framework of existing literature.

Response: We greatly appreciate the reviewer's positive appraisal and constructive comments. We have revised both "Introduction" and "Discussion" parts according to the reviewer's suggestion.

2. Various studies using PD-L1 and FasL for localized immunomodulation were not cited and discussed (PMID: 32814902; PMID: 36533683; PMID: 35559682; PMID: 35897156 PMID: 32923626; PMID: 32253240; PMID: 3659935 PMID: 31850658)

Response: The relevant studies from the references as the reviewer mentioned have been cited and discussed.

3. All flow cytometry data should include a gating strategy and the raw data in the form of dot plots should be presented either in the main figures or in the supplement. Without this information, it is difficult to appreciate the results.

Response: Many thanks for the reviewer's valuable suggestion. We have supplemented the gating strategy and dot plots in the Supplementary Information (Supplementary Figs 35-48).

4. Instead of plotting data from only n=3 samples and stating, 'representative of 2 or 3 experiments,' data pooled from independent studies (Fig. 2-5) should be presented to increase the sample numbers.

Response: We have increased the sample number to 6 for most of the experiments in Figs 2-5.

5. There are various inconsistencies in the main text and figure legends or materials and methods. For example, the result section refers to the quantification of FasL and PDL1 in MMV using a quantitative ELISA, while the supp figure 4b legend cites flow cytometry. Similarly, the main text in the results section mentions sorted T cells stimulated with ConA, while the supplement cites splenocytes being stimulated with ConA.

Response: Thanks for the reviewer's comments. We have updated the ELISA assay in the legend of Supplementary Fig. 4. In addition, Fig. 2d-f refer to the T cells within the Con A-treated splenocytes, while Fig 2k,m and Fig. 3g-j denote the sorted T cells stimulated with Con A. We have specified the cells used in the revised manuscript.

6. Fig. 2d. Apoptotic ratio should be defined, and the data should be presented in the form of dot plots.

Response: We have revised "Apoptotic ratio of the activated T cells" to "Total (early plus late) apoptotic percentages of T cells within the activated splenocytes" in the legend of Fig. 2d. The corresponding flow cytometric dot plots have been shown as Supplementary Fig. 6.

Revised Supplementary Figure 6. Apoptosis of T cells within the activated splenocytes after treatment with varying amounts of MMVs determined by Annexin V-FITC/PI double-staining assay. Representative is displayed from 6 independent experiments.

7. Fig. 2f. A blocking anti-PDL1 Ab results in higher levels of T cell proliferation as compared

with the culture without Ab. What is the contribution of FasL to the inhibition of proliferation? Also, the proliferation of cells without MMV should be shown as a positive control. Again, dot plots should be shown alongside bar graphs. Also, instead of the apoptotic ratio, the percent apoptosis of CD3 T cells over the background, i.e. MMV should be shown. Moreover, MMV from a control cell line without FasL or a blocking anti-FasL Ab should be used to assess the relative contribution of FasL and PDL1 to the observed T cell inhibition.

Response: As the reviewer suggested, we have performed the blocking experiment using the anti-PD-L1 and anti-FasL antibodies to evaluate the role of PD-L1 and FasL in inhibition of the T cell proliferation. The proliferation of cells without MMV treatment (the “Activated T cell” group) served as a positive control. The quantitative plots and flow cytometric histograms have been provided (Fig. 2f and Supplementary Fig. 9). Pre-treatment with the anti-PD-L1 antibody (aPD-L1) significantly restored the proliferation of the activated T cell that was inhibited by MMVs. In the FasL blockade assay, the proliferation of activated T cells was partially recovered. These results demonstrate that PD-L1 and FasL play essential and partial roles in the MMV-mediated suppression of activated T cell proliferation, respectively. We have revised apoptotic ratio to the percentage total apoptotic (early plus late apoptotic) for Fig. 2d. and the corresponding apoptosis dot plots have been provided in the Supplementary Information.

Revised Figure 2f. Proliferation of T cells within the activated splenocytes after treatment with MMVs in the absence and presence of aPD-L1 or aFasL determined by CFSE dilution assay. Data are shown as mean \pm s.d. ($n = 6$ independent samples). * $P < 0.05$, *** $P < 0.001$ (one-way ANOVA with Tukey post-hoc test).

Revised Supplementary Figure 9. Proliferation of T cells within the activated splenocytes after treatment with MMVs in the absence and presence of aPD-L1 or aFasL determined by CFSE dilution assay. Representative is displayed from 6 independent experiments.

8. Given the low percentage of the indicated cell populations ($\sim 0.2\%$), the gating strategy and dot plots for Fig. 31-n should be provided in the supplement. Additionally, there is an inconsistency between Fig. 31-n and supplementary Fig. 16 that shows a much higher T cell %, 18 for CD8⁺CD44-62L^{low}.

Response: The gate strategies have been supplemented in the Supplementary Information. The quantitative data (%) of Fig. 31-n and Supplementary Figs 19,22,23 (original Supplementary Figs 16-18) show the percentage of T_{eff} or T_{reg} cells in the whole kidney, DLN or spleen, while the flow cytometric analysis plots of Supplementary Figs 19,22,23 (original Supplementary Figs 16-18) present the cell population percentages in the CD8⁺ T or CD4⁺ T cells. We have added the relative descriptions in the legends of these figures.

9. Fig. 4e, f shows islet grafts with MMV-gel survive over 30 days and survival is maintained by Treg cells. Why were the animals only monitored for ~ 30 days given the excellent survival of the graft? Also, the injection of anti-C25 Ab to deplete Treg cells appears to cause immediate rejection, which is quite unusual per published literature in this model. One would expect a delayed rejection, >10 days after Ab injection. This point should be discussed.

Response: Thanks for the reviewer's comments. 66% of the islet/MMV-Gel-treated diabetic mice maintained normoglycemia for 30 d, while all mice receiving the islet and islet/L-Gel transplantation suffered from graft rejection within 14 and 18 d, respectively (Supplementary Fig. 24), indicating that islet/MMV-Gel significantly prolonged the survival of allogeneic islets. Thus, monitoring of BGL for 30 d is evident to demonstrate the efficacy of islet/MMV-Gel.

Previous researches have demonstrated that depletion of Treg cells by injection of anti-

CD25 antibody caused similar acute rejection of grafts (*Proc. Natl Acad. Sci. U. S. A.* 2019, 116, 13508-13516; *J. Immunol.* 2020, 204, 2840-2851). The speed of graft rejection was directly related to the depletion of Treg cells (*J. Immunol.* 2011, 187, 5901-5909). The high efficiency of Treg depletion in mice resulted in the rapid graft rejection of our results. If the mice have an inefficient depletion rate of Treg cells may lead to delayed rejection. In addition, several researches suggest that depletion of Treg cells by injection of anti-CD25 antibody may interfere with the Teff cell function (*Nat. Immunol.* 2007, 8, 191-197; *PLoS Pathog.* 2014, 10, e1003913). To address this issue, we have used the FoxP3/DTR mice that express the human DTR in the Treg cells. To assess that Treg cells were also responsible for enhanced graft survival, the islet/MMV-Gel-implanted FoxP3/DTR diabetic mice were treated by two i.p. injections of DT (50 ng/g) to directly deplete the Treg cells, followed by monitoring the BGL variation. Early treatment with DT at Day 14 and 15 led to rejection of all grafts within the following 18 d (Supplementary Fig. 25). The islet/MMV-Gel-transplanted diabetic mice receiving the DT treatment at Day 30 and 31 also experienced similar graft rejection, indicating that depletion of Treg cells causes rejection of graft under the established immune acceptance transplantation microenvironment (Fig. 4f). By comparison, the normal FoxP3/DTR mice treated with DT did not show any noticeable increase of BGLs that was in the range of normoglycemia (Supplementary Fig. 26). The results substantiate that the Treg cells play an essential role on allograft acceptance for the mice receiving the FasL/PD-L1-presenting MMV-Gel.

Revised Supplementary Figure 25. Changes in the individual BGLs of the islet/MMV-Gel-transplanted FoxP3/DTR diabetic mice after i.p. injection of DT ($n = 6$ mice). Black arrows indicate two i.p. injections of DT at Day 14 and 15.

Revised Figure 4f. Changes in the individual BGLs of the islet/MMV-Gel-transplanted FoxP3/DTR diabetic mice after i.p. injection of DT ($n = 6$ mice). Black arrows indicate two i.p. injections of DT at Day 30 and 31.

Revised Supplementary Figure 26. Changes in the individual BGLs of the FoxP3/DTR mice after i.p. injection of DT ($n = 6$ mice). Black arrows indicate two i.p. injections of DT at Day 0 and 1.

10. Long-term survival data was shown for the cohort transplanted with MMV-gel and rapamycin (Fig. 4i-j). Rapamycin was introduced without a rationale. If MMV-gel is sufficient on its own to prevent islet graft rejection, why use rapamycin? Also, Data with individual agents, i.e. MMV without IFN- γ treatment should be shown for comparison.

Response: With demonstration of 66% of mice transplanted with islet/MMV-Gel maintaining normoglycemia for 30 d, we further evaluated whether the survival of the allografts could be improved by combined treatment with low dosage of RAPA. RAPA as the first-line non-steroidal immunosuppressive drug was used for reducing the immune pressure on the allograft in the early post-transplantation period. Combination treatment with RAPA notably prolonged

the survival of allogeneic islet with MST of 16 d. In sharp contrast, 80% of islet/MMV-Gel survived within 30 d and 60% remained alive up to 100 d under the transient cover of RAPA (15 i.p. injections, once a day).

We have monitored the individual BGLs of the diabetic mice after transplantation of the islet-embedded gel crosslinked by MMVs without the IFN- γ treatment (islet/untreated-MMV-Gel) (Response Fig. 9). 44% of the mice implanted with islet/untreated-MMV-Gel retained BGLs below 200 mg/dL within 30 d, suggesting that pre-treatment of MMVs with IFN- γ to upregulate the surface expression of PD-L1 contributes to prolonged survival of allografts.

Response Figure 9. Changes in the individual BGLs of the diabetic mice after transplantation of islet/untreated-MMV-Gel ($n = 9$ mice).

11. Islet transplantation was performed in C57BL/6-to-BALB/c, which is a low responder, and not all controls reject the graft per published literature (PMID: 19364073). Testing the concept in BALB/c-to-C57BL/6, a high responder, will reinforce the drawn conclusion, but not required.

Response: According to the reviewer's constructive comments, we have evaluated the capability of MMV-Gel to provide immunoprotection for the islets from the BALB/c mice against attack by the activated T cells from the C57BL/6 mice by static GSIS assay. The hydrogel embedding the islets (BALB/c) was incubated with the activated T cells (C57BL/6) for 48 h. Exposed to both the first low and high concentration of glucose, the insulin secreted by islet/MMV-Gel showed significantly higher than that of islet/L-Gel (Response Fig. 10a). In the second phase of low glucose, compared with islet/L-Gel that was unable to switch off insulin secretion, islet/MMV-Gel functioned by responding to glucose variation. The stimulation index of islet/MMV-Gel was calculated to be significantly higher than that of islet/L-Gel (Response Fig. 10b). These results indicate the immunoprotective effect of MMV-Gel in a strong rejection

model (BALB/c to C57BL/6).

Response Figure 10. GSIS (a) and stimulation index (b) of islet/L-Gel and islet/MMV-Gel after incubation with the activated T cells for 48 h. The islets were isolated from the BALB/c mice, while the T cells were from extracted from the C57BL/6 mice. Data are shown as mean \pm s.d. ($n = 6$ independent samples). $P > 0.05$ (n.s.), $*P < 0.05$, $**P < 0.01$ (two-tailed unpaired t -test).

REVIEWERS' COMMENTS

Reviewer #1 (Remarks to the Author):

Thank you for addressing most of my points. I have a few remaining concerns as indicated below:

- In the introduction revise 'For example, the patients with type 1 diabetes are at high risk of recurrent autoimmunity and alloimmunity after islet transplantation, leading to allorejection with limited long-term benefit of the graft' with 'In patients with type 1 diabetes, transplanted islets are also targeted by the recurrence of autoimmunity which further limits long-term graft survival'
- Fig.36 and 37 for gating strategy should include the negative controls used for gating (IFNg not treated and/or unstained cells and/or isotype control)
- Data presented in the rebuttal in the response figures are important and should be included in main or supplementary figure and described in the main result section of the manuscript

Reviewer #2 (Remarks to the Author):

The authors addressed raised issues and revised the manuscript accordingly.

Reviewer #1:

Thank you for addressing most of my points. I have a few remaining concerns as indicated below:

- In the introduction revise ‘For example, the patients with type 1 diabetes are at high risk of recurrent autoimmunity and alloimmunity after islet transplantation, leading to allorejection with limited long-term benefit of the graft’ with ‘In patients with type 1 diabetes, transplanted islets are also targeted by the recurrence of autoimmunity which further limits long-term graft survival’

Response: Thanks for the reviewer’s valuable comments. We have revised this sentence as the reviewer suggested.

- Fig.36 and 37 for gating strategy should include the negative controls used for gating (IFN γ not treated and/or unstained cells and/or isotype control)

Response: We have included the negative control in the gating strategy for flow cytometric analysis in Supplementary Figs 38,39 (original Supplementary Figs 36,37).

Revised Supplementary Figure 38. Gating strategy for flow cytometric analysis on the expression of FasL (a,b) and PD-L1 (c,d) on the untreated and IFN- γ -treated MSCs (Fig. 2a, Fig. 2b and Supplementary Fig. 2).

Revised Supplementary Figure 39. Gating strategy for flow cytometric analysis on the expression of FasL on the surface of RBCs (a), PLTs (b) and aPLTs (c) (Fig. 2a).

- Data presented in the rebuttal in the response figures are important and should be included in main or supplementary figure and described in the main result section of the manuscript

Response: Key response figures have been included in the Supplementary Information, and the corresponding results have been described in the main text.

Reviewer #2:

The authors addressed raised issues and revised the manuscript accordingly.

Response: Many thanks to the reviewer for the contribution to the review.